

**Contrasting terrestrial carbon cycle responses to the two strongest El**
**Niño events: 1997-98 and 2015-16 El Niños**
Jun Wang[1,2], Ning Zeng[2,3], Meirong Wang[4], Fei Jiang[1], Hengmao Wang[1], and Ziqiang
Jiang[1]
[1]International Institute for Earth System Science, Nanjing University, Nanjing, China
[2] State Key Laboratory of Numerical Modelling for Atmospheric Sciences and Geophysical Fluid
Dynamics, Institute of Atmospheric Physics, Beijing, China
[3]Department of Atmospheric and Oceanic Science and Earth System Science Interdisciplinary
Center, University of Maryland, College Park, Maryland, USA
[4]Collaborative Innovation Center on Forest and Evaluation of Meteorological Disasters/Key
Laboratory of Meteorological Disaster of Ministry of Education, Nanjing University of
Information Science & Technology, Nanjing, China
Correspondence to: J. Wang (wangjun@nju.edu.cn)
**Abstract**
The large interannual $CO_2$ variability is dominated by the response of terrestrial
biosphere to El Niño-Southern Oscillation (ENSO). However, behaviors of terrestrial
ecosystems differ in patterns and biological processes in different El Niño events. Here
we conduct a comprehensive comparison of the two strongest El Niño events in history,
namely, the recent 2015-16 event, and the earlier 1997-98 event in the context of multi-
event 'composite' El Niño. We analyze Mauna Loa $CO_2$ concentration, surface carbon
fluxes from three atmospheric inversions, and a mechanistic carbon cycle model
VEGAS. We find large differences in the carbon cycle responses, even though the two
El Nino events are of similar magnitude.
We find that the land-atmosphere carbon flux ($F_{TA}$) anomaly in 1997-98 El Niño was




1.95 Pg C yr$^{-1}$ globally, but two times smaller during 2015-16 El Niño at 0.79 Pg C yr$^{-}$
$^{1}$. We also find that $F_{TA}$ had no obvious lagged response in 2015-16 El Niño, in contrast
to that in 1997-98 El Niño. Separating the global flux by major geographical regions,
during 1997-98, the fluxes in the tropics and extratropical northern hemisphere were
1.98 and $-0.04$ Pg C yr$^{-1}$, respectively. During 2015-16, these were 1.07 and $-0.4$ Pg
C yr$^{-1}$. Analysis of the mechanism shows that in the tropics, the widespread drier and
warmer conditions caused the decrease in gross primary productivity (GPP, $-1.11$ Pg
C yr$^{-1}$) and increase in terrestrial ecosystem respiration (TER, 0.49 Pg C yr$^{-1}$) in
1997-98 El Niño. During 2015-16, in contrast, anomalously wet conditions occurred in
Sahel and East Africa that caused increase in GPP, compensating its decrease over other
tropical regions. As a result, the total 2015-16 tropical GPP and TER anomalies were
0.29 and 1.23 Pg C yr$^{-1}$. GPP dominance during 1997-98 and TER dominance during
2015-16 accounted for the phase difference in their $F_{TA}$. In extratropical northern
hemisphere, we find that temperature was warmer both in 1997-98 and 2015-16 El
Niños over North America, contributing to enhancements in GPP and TER. However,
temperature over Eurasia was warmer in 2015-16 El Niño, opposing to the cooler in
1997-98 and composite El Niño events. This warmer condition enhanced GPP and TER
over the Eurasia in 2015-16 El Niño, compared to their suppressions in 1997-98 El
Niño. The total extratropical northern hemisphere GPP and TER anomalies were 0.86
and 0.74 Pg C yr$^{-1}$ in 1997-98 El Niño and 1.8 and 1.47 Pg C yr$^{-1}$ in 2015-16 El Niño.
Additionally, we find that wildfires played less important roles in 2015-16 El Niño than
in 1997-98 El Niño.





## 1 Introduction

The atmospheric $CO_2$ growth rate has a significant interannual variability, greatly influenced by the El Niño-Southern Oscillation (ENSO) (Bacastow, 1976; Keeling et al., 1995). This interannual variability primarily stems from terrestrial ecosystems (Bousquet et al., 2000; Zeng et al., 2005). Further, there is a general consensus that the tropical terrestrial ecosystems account for the terrestrial carbon variability (Cox et al., 2013; Peylin et al., 2013; Wang et al., 2016; Wang et al., 2013; Zeng et al., 2005). They tend to anomalously release C flux during El Niño episodes, and uptake during La Nina episodes (Wang et al., 2016; Zeng et al., 2005). Recently, Ahlstrom et al. (2015) further suggested that ecosystems over the semi-arid regions dominated the terrestrial carbon interannual variability with its 39% contribution.

The terrestrial dominance primarily results from the drive-response mechanisms in climate variability (especially in temperature and precipitation) caused by ENSO and plants/soil physiology (Jung et al., 2017; Tian et al., 1998; Wang et al., 2016; Zeng et al., 2005). Directly, land-atmosphere C flux ($F_{TA}$) is mainly attributable to the imbalance between the gross primary productivity (GPP) and terrestrial ecosystem respiration (TER), according to $F_{TA} = TER - GPP + C_{fire}$ ($C_{fire}$ is generally much smaller than GPP or TER). So variations in each of them or both result in the variations in $F_{TA}$.

Based on a dynamical global vegetation model (DGVM), Zeng et al. (2005) pointed out that net primary productivity (NPP) contributed to almost three fourth of the tropical $F_{TA}$ interannual variability. Later, multi-model simulations involved in TRENDY project consistently suggested NPP or GPP dominated the terrestrial carbon variability (Ahlstrom et al., 2015; Piao et al., 2013; Wang et al., 2016).

These biological process analyses inferred that precipitation variation was the dominant



climate factor in controlling $F_{TA}$ interannual variability (Ahlstrom et al., 2015; Qian et
al., 2008; Tian et al., 1998; Wang et al., 2016; Zeng et al., 2005). Quantitatively, Qian
et al. (2008) illustrated the contributions of the tropical precipitation and temperature
were 56 and 44% respectively, based on the model sensitivity experiments. Eddy
covariance network observations suggested that interannual C flux variability over the
tropical and temperate regions was controlled by precipitation, while boreal ecosystem
C fluxes were more subject to temperature and radiation (Jung et al., 2011). At the same
time, there was a significant positive correlation between atmospheric $CO_2$ growth rate
and mean tropical land temperature (Anderegg et al., 2015; Cox et al., 2013; Wang et
al., 2013; Wang et al., 2014). Sensitivity analysis indicated an about 3.5 Pg C yr$^{-1}$
anomaly in $CO_2$ growth rate with a 1℃ increase in tropical land temperature, whereas
only a weaker interannual coupling existed between $CO_2$ growth rate and tropical land
precipitation (Wang et al., 2013). Therefore, these studies (Anderegg et al., 2015; Cox
et al., 2013; Wang et al., 2013; Wang et al., 2014) suggested the temperature dominance
in $F_{TA}$ or $CO_2$ growth rate interannual variations, considering this strong emergent
linear relationship. Recently, in order to reconcile these contradictory reports, Jung et
al. (2017) illustrated that temporal and spatial compensatory effects in water availability
linked yearly global $F_{TA}$ variability to temperature.
Apart from these long-term time series studies on the interannual $F_{TA}$ or $CO_2$ growth
rate variability, we should keep in mind that the response of terrestrial carbon cycle to
every El Niño/La Nina event has its unique behaviors such as in the strength, spatial
pattern, biological process, and so on (Schwalm, 2011). For example, the wildfires
played an important role in $F_{TA}$ anomalies during 1997-98 El Niño (van der Werf et al.,
2004). Recently, one of the three strongest El Niño events in recorded history occurred
in  2015-16  years  (https://www.esrl.noaa.gov/psd/enso/current.html).  Given  the





disturbance of the El Chichón eruption in 1982-83 El Niño episode, we here attempt to
comprehensively compare the responses of terrestrial ecosystems to the two strongest
El Niños in 1997-98 and 2015-16 years in the context of multi-event 'composite' El
Nino, based on DGVM VEGAS in its Near-Real Time framework, inversion datasets
(CAMS, MACC, and CarbonTracker) and so on. Our purpose is to clarify the
distinctions in responses of biological processes in these two extreme events.
This paper is organized as follows: Section 2 describes the mechanistic carbon cycle
model used, its drivers, and reference datasets. Section 3 presents the results about the
total terrestrial C flux anomalies and spatial patterns along with their mechanisms.
Finally, discussions and concluding remarks are illustrated in Sect. 4.

## 113    2   Model and datasets

### 114    2.1 Mechanistic carbon cycle model and its drivers

In this study, we used the state-of-the-art VEGAS version 2.4 in its Near-Real Time
framework to investigate the responses of terrestrial ecosystems to El Niño events.
VEGAS has been widely used to study the terrestrial carbon cycle on its seasonal cycle,
interannual variability, and long-term trend (Zeng et al., 2005; Zeng et al., 2004; Zeng
et al., 2014). And it extensively participated in the international carbon modelling
project, such as the Coupled Climate-Carbon Cycle Model Intercomparison Project
(C[4]MIP) (Friedlingstein et al., 2006), TRENDY project (Sitch et al., 2015) and Multi-
scale Synthesis and Terrestrial Model Intercomparison Project (MsTMIP) (Huntzinger
et al., 2013). The detailed descriptions on its model structure, biological processes, and
so on can be referred to the appendix in Zeng et al. (2005). We ran VEGAS on the
0.5°×0.5° horizontal resolution from 1901 till the end of year 2016, and focus on the
period from 1980 to 2016.



The climate fields used to force VEGAS are as follows: (1) Precipitation datasets are
generated by combining the Climatic Research Unit (CRU) Time-series (TS) Version
3.22 (University of East Anglia Climatic Research Unit et al., 2014) , NOAA's
PRECipitation REConstruction over Land (PREC/L) (Chen et al., 2002), and NOAA
NCEP climate anomaly monitoring system-outgoing longwave radiation precipitation
index (CAMS-OPI) (Janowiak and Xie, 1999). (2) Temperature is adopted from the
CRU TS3.22 before the year 2013, and generated by combining CRU 1981-2010
climatology and the Goddard Institute for Space Studies (GISS) Surface Temperature
Analysis (GISTEMP) (Hansen et al., 2010) after 2013. (3) Downward shortwave
radiation is retrieved from the driver datasets in MsTMIP (Wei et al., 2014) before the
year 2010, and repeated the value of the year 2010 after it. Additionally, the gridded
cropland and pasture land use datasets are integrated from the History Database of the
Global Environment (HYDE) (Klein Goldewijk et al., 2011) with an linear
extrapolation in 2016.

**2.2 Reference datasets**
We here take a series of reference datasets as a comparison with the simulation of
VEGAS. The atmospheric $CO_2$ concentrations are from the monthly in-situ $CO_2$
datasets at Mauna Loa Observatory, Hawaii (Keeling et al., 1976). The Niño 3.4
(120°W–170°W, 5°S–5°N) sea surface temperature anomalies (SSTA) are adopted from
the NOAA Extended Reconstructed Sea Surface Temperature (ERSST), version 4
(Huang et al., 2015), with a 3-month running average. We take Copernicus Atmosphere
Monitoring Service (CAMS, 1980–2015), Monitoring atmospheric composition &
climate (MACC, 1980–2014) inversion results (Chevallier, 2013), and
CarbonTracker2016 (200001–201512) with the CarbonTracker Near-Real Time results



in 2016 (Peters et al., 2007) to compare with VEGAS. Fire emissions come from the
Global Fire Emissions Database, Version 4 (GFEDv4) from 1997 through 2014
(Randerson et al., 2015). Owing to the high correlation between solar-induced
chlorophyll fluorescence (SIF) and terrestrial GPP (Guanter et al., 2014), we take the
monthly satellite SIF from the GOME2_F version 26 from 2007 till 2016 (Joiner et al.,
2012). Another, we adopt the Enhanced Vegetation Index (EVI) from MODIS
MOD13C2 (Didan, 2015) to compare with the simulated leaf area index (LAI)
anomalies.
In order to get the anomalies during the El Niño events, we first remove the long-term
climatology in each dataset for getting rid of seasonal cycle signals, and then detrend
them based on the linear regression, because the trend is not caused by the interannual
variability.

**3   Results**
**3.1 Total terrestrial C flux anomalies**
Three strongest El Niño events (1982-83, 1997-98, and 2015-16) occurred from 1980
to 2016 with their maximum SST anomalies above 2.0 (Fig. 1a). El Niño event tends
to make the atmospheric $CO_2$ growth rate anomalously increase (Fig. 1b), so there are
two significant anomalously increased $CO_2$ growth rate corresponding to 1997-98 and
2015-16 El Niño events. Though the maximum increase in 2015-16 is a little smaller
than that in 1997-98. Owing to the diffuse light disturbance (Mercado et al., 2009) of
the eruption of Mount. El Chichón during 1982-83 El Niño event on the canonical
coupling between $CO_2$ growth rate anomalies and El Niño events, we mainly focus on
1997-98 and 2015-16 El Niño events in this study. The interannual variability of
atmospheric $CO_2$ growth rate principally originates from the terrestrial ecosystems (Fig.



1c). The correlation coefficient between $CO_2$ growth rate anomalies and global $F_{TA}$
simulated by VEGAS is 0.64 ($p < 0.05$). In order to evaluate the performance of
VEGAS simulation on the interannual time scale, we at the same time present CAMS,
MACC and CarbonTracker inversion results. We find that CAMS and MACC
inversions are nearly the same, both having the correlation coefficient about 0.60 ($p <$
$0.05$) with VEGAS. From 2000 through 2016, CarbonTracker is highly correlated with
VEGAS ($r = 0.71$, $p < 0.05$). These high correlation coefficients between VEGAS and
reference datasets underscore that VEGAS can well capture the terrestrial carbon cycle
interannual variability.
There are altogether 10 El Niño events from 1980 through 2016, each with different
duration and strength (Table 1). According to El Niño definition, we can find that these
10 El Niño events can be categorized into 2 weak (with a 0.5 to 0.9 SSTA), 3 moderate
(1.0 to 1.4), 2 strong (1.5 to 1.9), and 3 very strong (≥2.0) events. In 1997-98 El Niño,
the positive SSTA lasted from April 1997 to June 1998, while positive SSTA happened
in winter 2014, and extended to June 2016 in 2015-16 El Niño (Fig. 2a). However,
every El Niño event always peaks in winter (November or December) (Fig. 2a).
Considering this phase-lock phenomenon in El Niño events, we make a composite
analysis (getting rid of 1982-83 and 1991-92 because of the diffuse radiation
disturbances) as the background responses of terrestrial carbon cycle to El Niño events.
We can easily find that evolutions of $F_{TA}$ anomalies in VEGAS, mean of CAMS and
MACC, and CarbonTraker in composite, 1997-98, and 2015-16 El Niño events are
closely consistent with Mauna Loa CGR anomalies (Fig. 2b, c, and d). Peaks of $F_{TA}$
and Mauna Loa CGR anomalies in 1997-98 and 2015-16 El Niños are much stronger
than those in composite analysis. Importantly, there were significant terrestrial lagged
responses in composite and 1997-98 El Niño events, with the peak of $F_{TA}$ anomaly in





March to April in El Niño decaying year (Fig. 2b and c), consistent with previous
studies (Qian et al., 2008; Wang et al., 2016). But this lagged terrestrial response
disappeared in Mauna Loa CGR, VEGAS and CarbonTracker in 2015-16 El Niño (Fig.
2d). Further, in June 2016, the $F_{TA}$ anomaly of VEGAS and CarbonTracker significantly
dropped (sign changed), but Mauna Loa CGR dropped a little (no sign changed) (Fig.
2d). Similar phenomenon also occurred earlier from April to July 2015. In addition, we
can know that the anomalous C release caused by El Niño lasts about from July in the
El Niño developing year to October in the El Niño decaying year (Fig. 2b, c, and d).
For simplicity, we calculate the total anomalies in next context during this period for
all El Niño events, taking the terrestrial lagged responses into account (Wang et al.,

212   2016).

According to major geographical regions, we separate global $F_{TA}$ anomaly into
extratropical northern hemisphere (23°N–90°N), tropical regions (23°S–23°N), and
extratropical southern hemisphere (60°S–23°S). Because $F_{TA}$ anomaly over the
extratropical southern hemisphere is generally smaller, we mainly present the
evolutions of $F_{TA}$ over the extratropical northern hemisphere and tropical regions in Fig.
3. Comparing the global and tropical $F_{TA}$ anomalies, we find that $F_{TA}$ anomalies in
tropical regions dominate the global $F_{TA}$ in these events (Fig. 3b, d and f), in accord
with previous conclusions (Peylin et al., 2013; Zeng et al., 2005). Additionally, $F_{TA}$
anomalies over the extratropical northern hemisphere are nearly neutral in VEGAS
during composite and 1997-98 El Niño events (Fig. 3a and c). But we find that there
were obvious anomalous uptakes from April to September in 2016 simulated by
VEGAS (Fig. 3e), compensating the release over the tropics (Fig. 3f). These anomalous
uptakes made the global negative $F_{TA}$ anomalies from May to September in 2016 (Fig.
2d). Similar anomalous uptake happened over the extratropical northern hemisphere



earlier from April to July 2015. These anomalous uptakes in VEGAS are to some extent
consistent with results in CarbonTracker, and well account for the global $F_{TA}$ drops
mentioned above in these periods. Comparing the behaviors between Mauna Loa CGR
and $F_{TA}$ anomalies, we can now clearly find that Mauna Loa CGR, coming from the
tropical observatory, does not reflect the signals over the extratropical northern
hemisphere in time (Fig. 2d and Fig. 3e).
Because $F_{TA}$ mainly stems from the difference between TER and GPP, we present TER
and GPP anomalies in Fig. 4 in order to well explain the $F_{TA}$ anomalies. We find that
anomalous negative GPP dominated the $F_{TA}$ anomaly in tropics during composite and
1997-98 El Niño episodes with the significant lagged responses (peak at about May in
El Niño decaying year) (Fig. 4b and d). Besides, obvious positive TER anomalies
occurred from October 1997 to April 1998 (Fig. 4d), contributing to tropical C release
in this period (Fig. 3d). In contrast, we find that anomalous positive TER dominated
the $F_{TA}$ anomaly in tropics during 2015-16 El Niño episode without obvious lags (Fig.
4f), accounting for the disappearance of terrestrial $F_{TA}$ lagged response (Fig. 2d). In the
extratropical northern hemisphere, increased GPP and TER from April to October in
composite and 1998 were nearly identical (Fig. 4a and c), making neutral $F_{TA}$ anomalies
(Fig. 3a and c). But increased GPP was stronger than increased TER from April to July
2015 and from April to September 2016 (Fig. 4e), resulting in the anomalous uptake in
$F_{TA}$ (Fig. 2d and Fig. 3e).
Quantitatively, we calculated the total C flux anomalies from July in El Niño
developing year till October in El Niño decaying year. The composite global $F_{TA}$
anomaly during El Niño events in VEGAS is about 0.71 Pg C $yr^{-1}$, dominated by
tropical ecosystems with 0.74 Pg C $yr^{-1}$ (Table 2). These anomalies are comparable to
the mean of CAMS and MACC inversion results with $0.92 \pm 0.01$ globally and



$0.66\pm0.03$ Pg C yr$^{-1}$ in tropics. In these two extreme cases, a very strong anomalous C
release occurred in 1997-98 El Niño episode with a value of 1.93 Pg C yr$^{-1}$, close to
2.57 Pg C yr$^{-1}$ in CAMS and MACC inversions, while only 0.79 Pg C yr$^{-1}$ was released
in 2015-16 El Niño episode, comparable to 0.82 Pg C yr$^{-1}$ in CarbonTracker. But $F_{TA}$
anomalies in tropical regions dominated the global $F_{TA}$ anomalies in both cases with
respective values of 1.98 and 1.07 Pg C yr$^{-1}$ in VEGAS. Moreover, anomalous C uptake
simulated by VEGAS over the extratropical northern hemisphere cancelled 0.40 Pg C
yr$^{-1}$ anomalous release in tropics in 2015-16 El Niño, while it was neutral ($-0.04$ Pg C
yr$^{-1}$) in 1997-98 El Niño. And the $F_{TA}$ anomaly was relatively smaller in the
extratropical southern hemisphere.
In biological processes, GPP ($-1.11$ Pg C yr$^{-1}$) and TER (0.49 Pg C yr$^{-1}$) in tropics
together drove the anomalous $F_{TA}$ in 1997-98, while TER (1.23 Pg C yr$^{-1}$) partly
cancelled by GPP (0.29 Pg C yr$^{-1}$) drove the anomalous $F_{TA}$ in 2015-16 (Table 2). These
data confirmed that GPP played the more important role in 1997-98, while TER
dominance occurred in 2015-16 El Niño episode. In the extratropical northern
hemisphere, GPP and TER cancelled each other. They had respective 0.20 and 0.12 Pg
C yr$^{-1}$ in composite analysis and 0.86 and 0.74 Pg C yr$^{-1}$ in 1997-98 El Niño, making
the nearly neutral $F_{TA}$ anomaly there. But GPP (1.80 Pg C yr$^{-1}$) was stronger than TER
(1.47 Pg C yr$^{-1}$) in 2015-16 El Niño, causing the significant C uptake. Additionally, $F_{TA}$
anomaly caused by wildfires also played an important role in 1997-98 El Niño episode
with globally 0.46 Pg C yr$^{-1}$ in VEGAS, consistent with GFED fire data product (0.82
Pg C yr$^{-1}$). The effect of wildfires on $F_{TA}$ anomaly in 1997-98 El Niño episode has been
suggested by van der Werf et al. (2004). But it was close to zero (0.08 Pg C yr$^{-1}$) in
2015-16 El Niño episode.

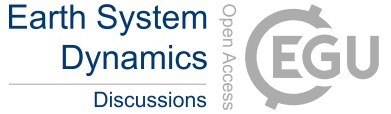

**3.2 Spatial features and its mechanisms**
Regional responses of terrestrial ecosystems to El Niño events are inhomogeneous,
principally according to the anomalies in climate variability. In composite El Niño
analysis (Fig. 5a), land consistently releases C flux in tropics, while it anomalously
uptakes C flux over North America as well as the central and eastern Europe. These
regional responses are generally consistent with the CAMS and MACC inversion
results (Fig. 5d).
In 1997-98 El Niño episode, tropical responses were analogous to composite results
except the stronger releases. North America and central and eastern China had stronger
C uptake, whereas Europe and Russia had stronger C release (Fig. 5b). However, in
2015-16 El Niño episode, anomalous C uptake happened over the Sahel and east Africa,
compensating the C release over the other tropical regions (Fig. 5c). It made the total
$F_{TA}$ anomaly in tropics in 2015-16 smaller than that in 1997-98 (Fig. 3d and f, and Table
2). North America had anomalous C uptake, similar to that in composite and 1997-98
El Niño, while central and eastern Russia also had anomalous C uptake in 2015-16 El
Niño (Fig. 5c), opposing to C release in composite and 1997-98 El Niño. This opposing
behavior of boreal forests over the central and eastern Russia clearly contributed to the
total uptake over the extratropical northern hemisphere (Table 2). Moreover, we can
clearly find that these regional responses in 2015-16 El Niño episode are significantly
consistent with the CarbonTracker result (Fig. 5f).
In order to better make the explanations on these regional C flux anomalies, we present
the main climate variabilities of soil wetness (mainly caused precipitation) and air
temperature, as well as the biological processes of GPP and TER in Fig. 6. In the
composite analyses, the soil wetness is generally reduced in tropics (Fig. 6a), making
the widespread decrease in GPP (Fig. 6b), verified by model sensitivity experiments



(Qian et al., 2008). At the same time, air temperature is anomalously warmer,
contributing to the enhancement in TER. But the drier conditions in the semi-arid
regions such as Sahel, South Africa, and Australia, restrict the enhancement in TER
induced by warmer temperature (Fig. 6d). Higher air temperature over the North
America largely enhances GPP and TER, while cooler conditions over the Eurasia will
reduce them (Fig. 6b–d). Wetter conditions over part of North America and Eurasia also
to some extent benefit GPP and TER (Fig. 6a).
Comparing the composite results (Fig. 6a–d) and 1997-98 El Niño episode (Fig.6e–h),
we can easily find that the regional patterns are almost identical except the difference
in magnitude. In contrast, there are some differences in 2015-16 El Niño episode. Over
the Sahel and East Africa, the soil wetness increased induced by more precipitation (Fig.
6i), dynamically making the air temperature cooler (Fig. 6k). This wetter condition
largely benefit GPP (Fig. 6j), compensating the decreased GPP over the other tropical
regions. It caused in total the increased GPP in tropics, opposing to composite and 1997-
98 El Niño episode (Table 2). More soil moisture also contributed to increase in TER
over the Sahel (Fig. 6l), contrary to that in 1997-98 El Niño episode (Fig. 6h). This
spatial compensation in GPP together with the widespread increased TER well
accounted for the TER dominance in tropics during 2015-16 El Niño episode. Besides,
increased GPP resulted in the anomalous C uptake here (Fig. 5c) which partly
compensated the anomalous C release over the other tropical regions. It in some degree
made the tropical smaller $F_{TA}$ in 2015-16 El Niño episode than that in 1997-98 El Niño
episode. Another obvious difference happened over the Eurasia with almost opposite
signals in 1997-98 and 2015-16 El Niño episodes. Air temperature during 2015-16 El
Niño episode over the Eurasia was anomalously higher, opposing to the cooler during
composite and 1997-98 El Niño (Fig. 6c, g, and k). This warmer condition enhanced





327 GPP and TER (Fig. 6j and l), contrary to their suppressions in composite and 1997-98

328 El Niño (Fig. 6b, d, f, and h). This phenomenon explained stronger GPP and TER

329 anomalies and anomalous C uptake over the whole extratropical northern hemisphere

330 (Table 2).

331 Recently, more attentions have been paid on SIF as an effective indicator for GPP

332 (Guanter et al., 2014). Therefore, we here try to make a comparison between simulated

333 GPP and SIF variabilities on the interannual timescale. Though there are noisy signals

334 in SIF, we can find that SIF was anomalously positive over the USA, part of Europe,

335 and East Africa, and negative over the Amazon and South Asia during the 2015-16 El

336 Niño episode, corresponding to the increased and decreased GPP respectively (Fig. 7a

337 and c). The correspondences over the other regions were not significant. In addition,

338 MODIS EVI anomalously increased over the America, Southern South America, part

339 of Europe, Sahel, and East Africa, but decreases over the Amazon, Northern Canada,

340 central Africa, South Asia, and Northern Australia (Fig. 7d). These EVI anomalies were

341 well corresponding to simulated LAI anomalies (Fig. 7b). These good correspondences

342 between simulated GPP (LAI) and SIF (EVI) give us more confidence in VEGAS

343 simulations.

344 At last, wildfires as important disturbances for $F_{TA}$ always release C flux. Though $F_{TA}$

345 anomalies caused by wildfires are generally smaller than GPP or TER anomalies, they

346 played an important role in 1997-98 El Niño episode (Globally 0.46 Pg C $yr^{-1}$ in

347 VEGAS and 0.82 Pg C $yr^{-1}$ in GFED) (Table 2), consistent with the previous study (van

348 der Werf et al., 2004). Here we show the $F_{TA}$ anomalies caused by wildfires in Fig. 8.

349 The correlation coefficients between simulated global $F_{TA}$ anomalies caused by

350 wildfires and GFED fire data product are 0.40 (unsmoothed) and 0.61 (smoothed) (Fig.

351 8a), confirming that VEGAS has certain capability in simulating this disturbance. In



1997-98 El Niño episode, satellite-based GFED data showed that $F_{TA}$ anomalies caused
by wildfires mainly happened over the tropical regions, such as Amazon, Central Africa,
South Asia, and Indonesia (Fig. 8d). VEGAS also simulated the positive $F_{TA}$ over these
tropical regions (Fig. 8b). The total tropical $F_{TA}$ anomalies caused by fires were 0.39
Pg C yr$^{-1}$ in VEGAS and 0.72 Pg C yr$^{-1}$ in GFED (Table 2). In 2015-16 El Niño episode,
wildfires also resulted in positive $F_{TA}$ anomalies over Amazon, South Asia, and
Indonesia, but their magnitudes were smaller than those in 1997-98 El Niño episode,
because it was much drier in 1997-98 El Niño episode than in 2015-16 El Niño episode
(Fig. 6e and i). In addition, the wetter conditions over the East Africa in 2015-16 El
Niño episode depressed the occurrences of wildfires with the negative $F_{TA}$ anomalies
(Fig. 8c). The tropical $F_{TA}$ anomaly in total was 0.13 Pg C yr$^{-1}$ in VEGAS (Table 2).
Therefore, we can find that wildfires played less important roles in 2015-16 than in
1997-98 El Niño episode. $F_{TA}$ anomalies caused by wildfires over the extratropics are
much weaker than those over the tropics, and their correspondences between VEGAS
and GFED are poorer (Table 2 and Fig. 8b and d).

**4   Conclusions and Discussions**
Climate anomalies in magnitudes and patterns caused by different El Niño events are
inconsistent, so responses of terrestrial ecosystems remain uncertain to different El
Niño events (Schwalm, 2011). In this study, we comprehensively compare the impacts
of the two strongest El Niño events in history, namely, the recent 2015-16, and earlier
1997-98 events in the context of multi-event 'composite' El Nino on the terrestrial
carbon cycle, relying on VEGAS in its Near-Real Time framework, inversion datasets
and so on. Main conclusions are drawn as follows:
(1) Simulations indicate that $F_{TA}$ anomaly in 2015-16 El Niño episode was globally





0.79 Pg C yr$^{-1}$, nearly two times smaller than that in 1997-98 El Niño (1.95 Pg C
yr$^{-1}$), confirmed by inversion results. We also find that $F_{TA}$ had no obvious lagged
response in 2015-16 El Niño, in contrast to that in 1997-98 El Niño. Separating the
global flux, we find that fluxes in the tropics and extratropical northern hemisphere
were 1.07 and $-0.4$ Pg C yr$^{-1}$ during 2015-16 El Niño episode respectively, while
these were 1.98 and $-0.04$ Pg C yr$^{-1}$ during 1997-98 event. Tropical $F_{TA}$ anomalies
dominated global $F_{TA}$ anomalies in both extreme El Niño events.
(2) Mechanistic analysis indicates that anomalously wetter conditions happened over
the Sahel and East Africa during 2015-16 El Niño episode, resulting in the increase
of GPP, which compensated the reduction of GPP over the other tropical regions. It
caused in total the increased GPP in tropics (0.29 Pg C yr$^{-1}$), opposing to composite
analysis ($-0.80$ Pg C yr$^{-1}$) and 1997-98 El Niño ($-1.11$ Pg C yr$^{-1}$). Spatial
compensation in GPP and widespread increased TER (1.23 Pg C yr$^{-1}$) well
explained the TER dominance in 2015-16 El Niño episode, opposing to GPP
dominance in 1997-98 event. Different biological dominance accounted for the
phase difference in $F_{TA}$ responses in 1997-98 and 2015-16 El Niños.
(3) Higher air temperature over the North America largely enhanced GPP and TER
both in 1997-98 and 2015-16 El Niño episodes. However, air temperature during
2015-16 El Niño episode over the Eurasia was anomalously higher, opposing the
cooler in 1997-98 El Niño episode. This warmer condition benefited GPP and TER,
well accounting for stronger GPP (1.80 Pg C yr$^{-1}$) and TER (1.47 Pg C yr$^{-1}$)
anomalies and anomalous C uptake ($-0.40$ Pg C yr$^{-1}$) over the extratropical
northern hemisphere during 2015-16 El Niño.
(4) Wildfires, frequently happening in tropics, played an important role in $F_{TA}$
anomalies during 1997-98 El Niño episode, confirmed by VEGAS simulation and



satellite-based GFED fire product. But VEGAS simulation indicates that the
tropical $F_{TA}$ caused by wildfires during 2015-16 El Niño episode was relatively
smaller than that during 1997-98 El Niño episode. This result was mainly because
the tropical weather was much drier in 1997-98 El Niño than that in 2015-16 El
Niño.

**Data Availability**
In this study, all the datasets can be freely accessed. Mauna Loa monthly $CO_2$ records
are available at https://www.esrl.noaa.gov/gmd/ccgg/trends/data.html. ERSST4
Niño3.4 index can be accessed from
http://www.cpc.ncep.noaa.gov/data/indices/ersst4.nino.mth.81-10.ascii. CAMS and
MACC inversions are available at http://apps.ecmwf.int/datasets/. CarbonTracker
datasets can be found at https://www.esrl.noaa.gov/gmd/ccgg/carbontracker/.
GFEDv4 global fire emissions are downloaded at https://daac.ornl.gov/cgi-
bin/dsviewer.pl?ds_id=1293. Satellite SIF datasets are retrieved from
http://avdc.gsfc.nasa.gov/pub/data/satellite/MetOp/GOME_F/MetOp-A/level3/.
MODIS enhanced vegetation index (EVI) datasets are downloaded from
https://lpdaac.usgs.gov/dataset_discovery/modis/modis_products_table/mod13c2_v00

420    6.


**Acknowledgements:**
We gratefully appreciate the ESRL for the use of their Mauna Loa atmospheric $CO_2$
records and CarbonTracker datasets, NOAA for ERSST4 ENSO index, LSCE-IPSL for
CAMS and MACC inversion datasets, the Oak Ridge National Laboratory Distributed
Active Archive Center for GFEDv4 global fire emissions, NASA Goddard Space Flight





Center for SIF datasets, and Land Processes Distributed Active Archive Center for
MODIS EVI datasets.

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



## Tables and Figures:

**Table 1** Lists of El Niño events from 1980 till 2016.

| No. | El Niño Events | Duration (mo) | Maximum Nino3.4 Index (°C) |
|---|---|---|---|
| 1 | Apr1982–Jun1983 | 15 | 2.1 |
| 2 | Sep1986–Feb1988 | 18 | 1.6 |
| 3 | Jun1991–Jul1992 | 14 | 1.6 |
| 4 | Oct1994–Mar1995 | 6 | 1.0 |
| 5 | May1997–May1998 | 13 | 2.3 |
| 6 | Jun2002–Feb2003 | 9 | 1.2 |
| 7 | Jul2004–Apr2005 | 10 | 0.7 |
| 8 | Sep2006–Jan2007 | 5 | 0.9 |
| 9 | Jul2009–Apr2010 | 10 | 1.3 |
| 10 | Nov2014–May2016 | 19 | 2.3 |

**Table 2** Carbon flux anomalies during El Niño events, calculated as the mean from July in the El Niño developing year to October in El Niño decaying year. Flux units are in Pg C yr$^{-1}$.

| Zones | El Niños | Inversions $F_{TA}$ (CAMS+MACC)[a] | Inversions $F_{TA}$ (CarbonTracker) | VEGAS Model $F_{TA}$ | VEGAS Model GPP | VEGAS Model TER | VEGAS Model $C_{fire}$ | GFED $C_{fire}$ |
|---|---|---|---|---|---|---|---|---|
| Global | composite[b] | 0.92±0.01 | – | 0.71 | −0.76 | −0.20 | 0.15 | – |
| | 1997-98 | 2.57±0.04 | – | 1.93 | −0.11 | 1.36 | 0.46 | 0.82 |
| | 2015-16 | – | 0.82 | 0.79 | 1.79 | 2.50 | 0.08 | – |
| NH | composite | 0.20±0.02 | – | −0.09 | 0.20 | 0.12 | −0.01 | – |
| | 1997-98 | 0.40±0.07 | – | −0.04 | 0.86 | 0.74 | 0.07 | 0.11 |
| | 2015-16 | – | 0.18 | −0.40 | 1.80 | 1.47 | −0.06 | – |
| Tropical | composite | 0.66±0.03 | – | 0.74 | −0.80 | −0.22 | 0.16 | – |



| | | | | | | | | |
|---|---|---|---|---|---|---|---|---|
| | 1997-98 | 2.12±0.14 | – | 1.98 | −1.11 | 0.49 | 0.39 | 0.72 |
| | 2015-16 | – | 0.53 | 1.07 | 0.29 | 1.23 | 0.13 | – |
| | composite | 0.07±0.01 | – | 0.06 | −0.16 | −0.10 | 0.00 | – |
| SH | 1997-98 | 0.05±0.02 | – | −0.01 | 0.13 | 0.13 | 0.00 | −0.01 |
| | 2015-16 | – | 0.11 | 0.12 | −0.31 | −0.19 | 0.00 | – |

[a]It represents the mean value of CAMS and MACC inversion results with the

uncertainty of their standard deviation.

[b]Composite analyses exclude 1982-83, 1991-92, and 2015-16 El Niños, because the

former two cases are disturbed by eruptions of El Chichón and Pinatubo, and the latter

is not covered by inversion datasets.

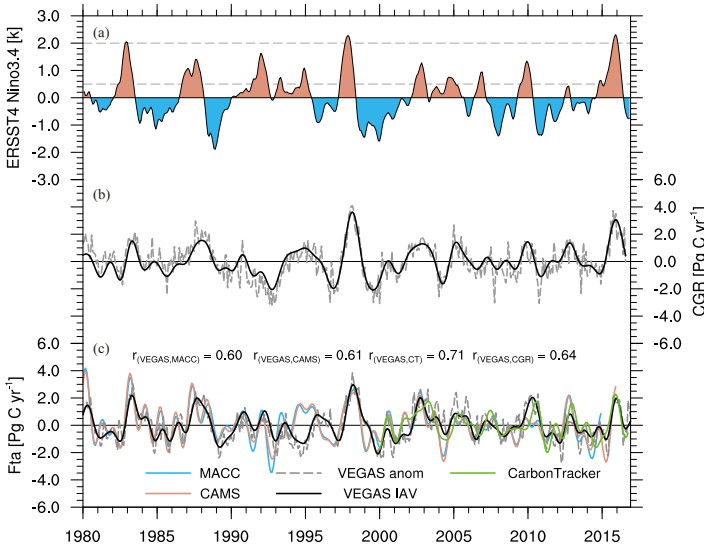

**Figure 1.** Interannual variabilities (IAV) in sea surface temperature anomaly (SSTA)

and carbon cycle. (a) ERSST4 Niño3.4 Index (units: K). It is the 3-month running

averaged SST anomaly for the Niño 3.4 region (5°N–5°S, 120°–170°W). (b) IAV in

MLO $CO_2$ growth rate (CGR, units: Pg C yr$^{-1}$). CGR is calculated as the difference



between the monthly mean in the adjacent years. The dashed line is the detrended
anomaly and the solid line is smoothed by the butterworth filtering. (c) IAV in land-
atmosphere carbon fluxes ($F_{TA}$, units: Pg C yr$^{-1}$). Blue and orange solid lines are the
smoothed results of MACC and CAMS inversions. Gray dashed line is the detrended
anomaly and black one is the smoothed result in VEGAS model simulation. The green
solid line is the smoothed CarbonTracker result.

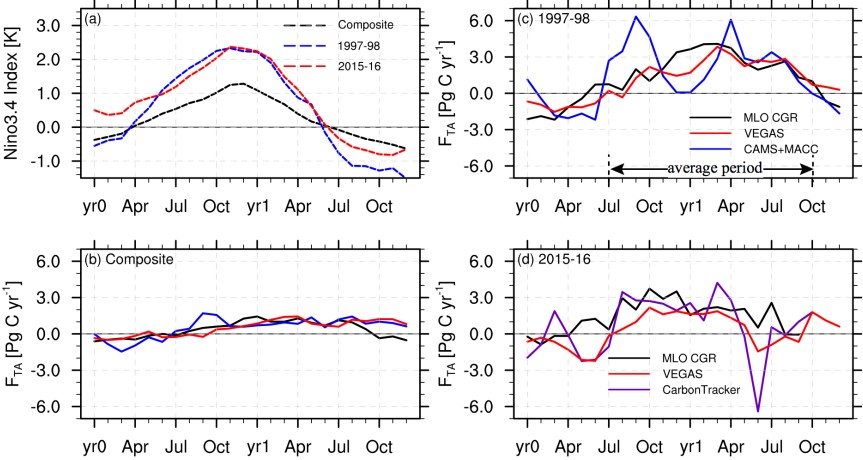


**Figure 2.** Evolutions of global $F_{TA}$ along with the development of El Niño. (a) shows
the SSTA in composite (in black), 1997-98 (in blue), and 2015-16 (in red) El Niño
events. (b) illustrate the $F_{TA}$ anomalies in El Niño composite analysis. The black solid
line denotes Mauna Loa CGR; red and blue lines show the VEGAS and mean of CAMS
and MACC inversions, respectively. (c) shows the $F_{TA}$ anomalies during 1997-98 El
Niño events. And the arrows demonstrate the time periods during which we calculate
the C flux anomalies in table and below space figures. (d) demonstrate the $F_{TA}$
anomalies during 2015-16 El Niño events. And the purple line denotes result of
CarbonTracker2016 and CarbonTracker Near Real-Time datasets.






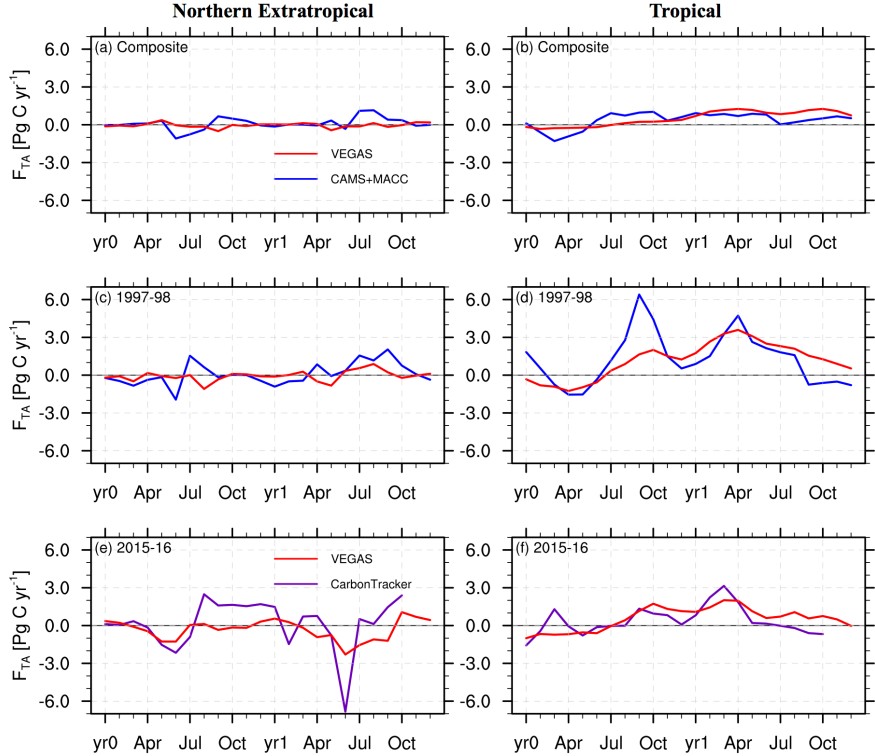


**Figure 3**. Evolutions of $F_{TA}$ over the extratropical northern hemisphere and tropical

regions along with the development of El Niño. (a–b) show the composite results with

VEGAS simulation (red solid line) and mean of CAMS and MACC inversions (blue

solid line). (c–d) show the $F_{TA}$ anomalies in 1997-98 El Niño. (e–f) demonstrate the

FTA anomalies in 2015-16 El Niño with VEGAS (red solid line) and CarbonTracker

(purple solid line).

619





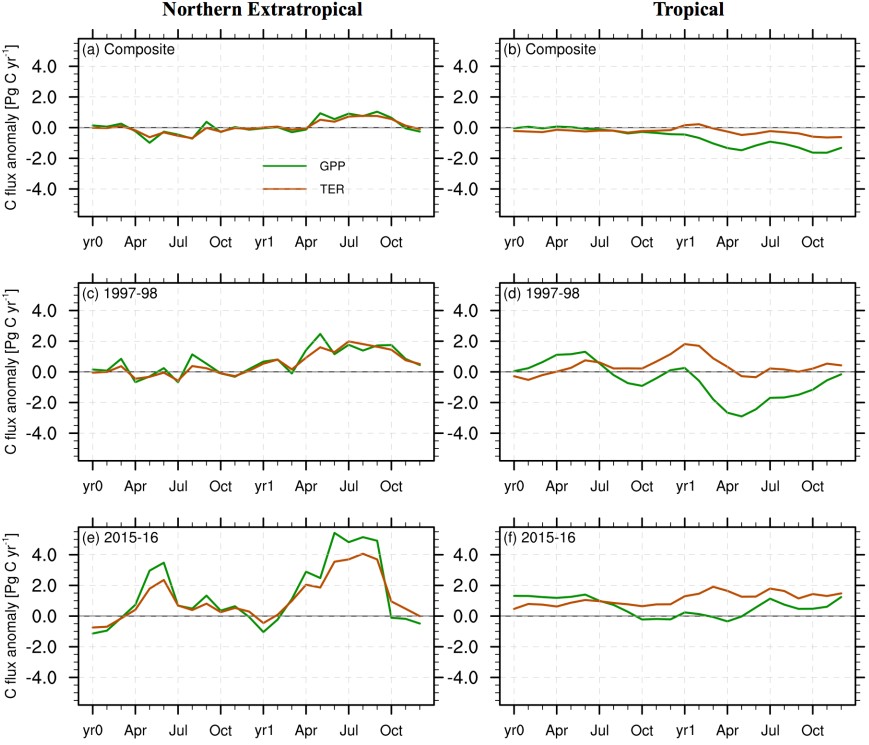

**Figure 4.** Evolutions of gross primary productivity (GPP, green lines) and terrestrial ecosystem respiration (TER, brown lines) over the extratropical northern hemisphere and tropical regions along with the development of El Niño. (a–b) show the El Niño composite results. (c–d) show the results in 1997-98 El Niño event. And (e–f) demonstrate the results in 2015-16 El Niño event.



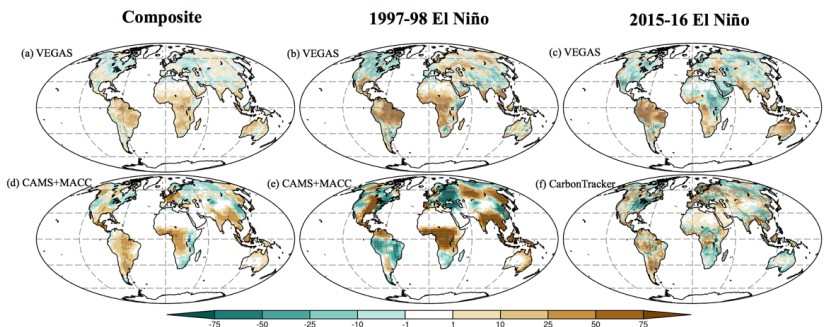

627

**Figure 5.** Spatial $F_{TA}$ anomalies calculated from July in El Niño developing year to

October in El Niño decaying year (units: Pg C yr$^{-1}$). (a), (b), and (c) show the results of

composite, 1997-98, and 2015-16 El Niño events simulated by VEGAS, respectively.

(d) and (e) represent the averaged results of CAMS and MACC in composite and 1997-

98 El Niños. (f) shows 2015-16 El Niño $F_{TA}$ anomaly in CarbonTracker.

633

634





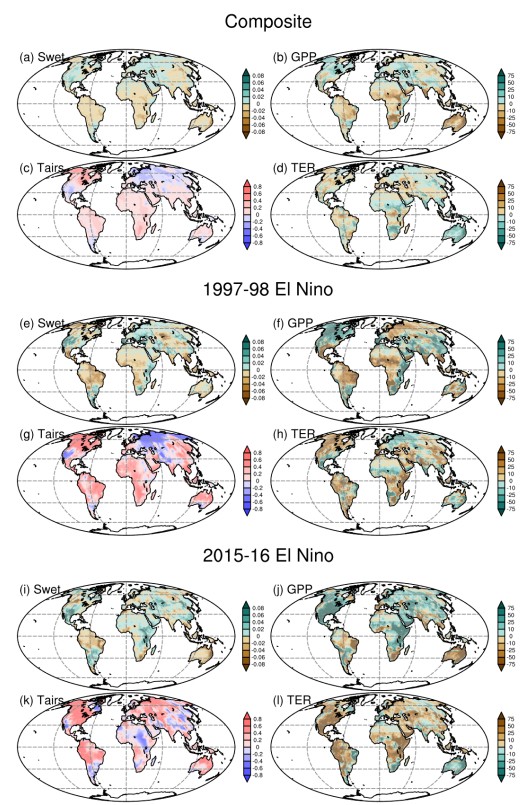

635

**Figure 6.** Anomalies in soil wetness, air temperature (units: K), gross primary productivity (GPP, g C m$^{-2}$ yr$^{-1}$), and terrestrial ecosystem respiration (TER, g C m$^{-2}$ yr$^{-1}$) from July in El Niño developing year to October in El Niño decaying year in composite, 1997-98, and 2015-16 El Niño episodes, respectively. (a–d) represent the results in composite analyses. (e–h) represent the anomalies during 1997-98 El Niño episode. And (i–l) show the anomalies during 2015-16 El Niño episode.

642





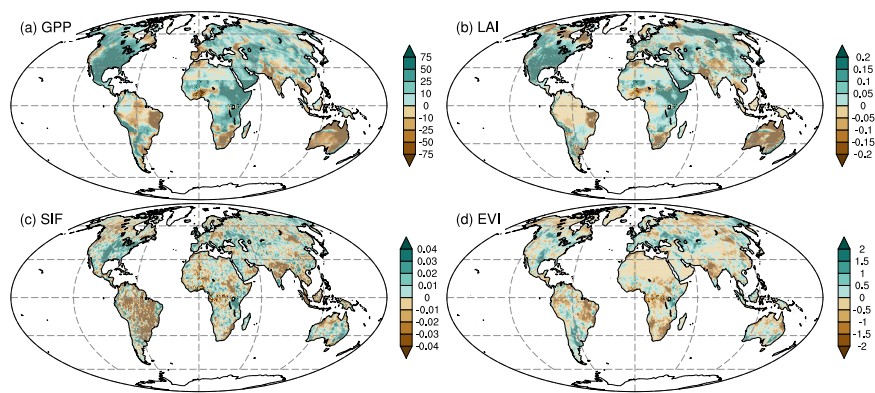

643

**Figure 7.** Spatial anomalies in (a) simulated GPP by VEGAS (units: g C m$^{-2}$ yr$^{-1}$), (b) simulated leaf area index (LAI, units: m$^2$ m$^{-2}$), (c) solar-induced chlorophyll fluorescence (SIF, units: mW m$^{-2}$ nm$^{-1}$ sr$^{-1}$), and (d) MODIS enhanced vegetation index (EVI, ×10$^{-2}$) from July 2015 to October 2016.

648

649

650

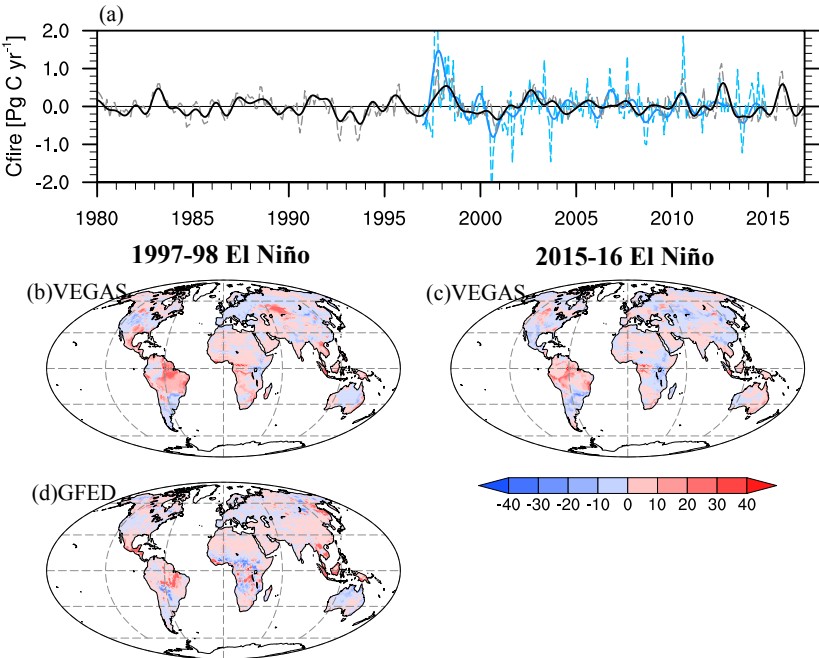

**Figure 8.** $F_{TA}$ anomalies induced by wildfires. (a) global total anomalies (Pg C yr$^{-1}$). The dashed gray and solid black lines represent the detrended and smoothed by butterworth filtering anomalies simulated by VEGAS. The dashed and solid blue lines represent the GFED results. (b) spatial $F_{TA}$ anomaly (g C m$^{-2}$ yr$^{-1}$) in 1997-98 El Niño episode in VEGAS. (c) spatial $F_{TA}$ anomaly in 2015-16 El Niño episode in VEGAS. (d) GFED anomaly in 1997-98 El Niño episode.