# Peer review of "Contrasting terrestrial carbon cycle responses to the two strongest El Niño events: 1997-98 and 2015-16 El Niños"

_Earth System Dynamics, 2017_

## Referee Comment (RC1) · Anonymous Referee #1 · 27 Jun 2017

General Comments:

This manuscript addresses the pertinent question of what terrestrial mechanisms and processes control inter-annual variability in atmospheric $CO_2$ growth rate during El Niño events, which is an area of research relevant to the scope of Earth System Dynamics. The title clearly reflects the research study and manuscript contents. The paper is a significant contribution to the field because there is large variability in carbon fluxes among models, especially during El Niño events, and it needs to be better understood. Through a multi-comparison study of model output, inversion studies and relevant datasets, this paper is a significant contribution to the study of earth system dy-

namics bridging the gap in our understanding of land-atmosphere carbon flux response to climate variability. The literature review is comprehensive and clearly identifies the current state of the science. The manuscript contents are organized in a logical manner. The experiment is well designed and sufficient to answer the questions posed in the manuscript. The model results are presented with an analysis of the mechanisms driving them, and the conclusions drawn are consistent with the interpretation of the results. The tables and figures are clear, include relevant information for the study and are organized in a logical manner.

Specific Comments:

1) Introduction: While the literature review is comprehensive and the introduction clearly describes the problem and the state of the science, the novelty of this research needs to be more clearly stated in the introduction. I suggest including a sentence explicitly stating how this research is novel compared to previous studies up front so the reader can better understand how this research is set apart from other studies. 2) Conclusions and Discussion: The conclusions are clearly outlined and are consistent with the interpretation of the results. However, this section seems to be more conclusion, and is lacking in discussion. This left me interested with many questions that should be added after the conclusions, such as the caveats of this study (model, datasets, etc.), implications of the research (i.e., how does this research advance our science), and what, if any, future research may be done to build on the conclusions established (i.e., additional model/data analysis, additional El Niño years analyzed, etc.). More discussion would tie the manuscript and the state of the science in better, and will give a better big picture view.

Technical Corrections:

1) Line 16: It is not clear what CO2 variability is being addressed. Perhaps, specify "The large interannual atmospheric CO2 variability. . ." 2) Line 21: Same comment as above, "Mauna Loa atmospheric CO2 concentration. . ." 3) Line 42: ". . .opposing to the

cooler in. . ." would read better as "opposing the cooling in. . ." 4) Line 68: for consistency and clarity, the variable "Cfire" should have a written definition included like the other variables, such as "carbon flux from fire". 5) Line 73: ". . .involved in TRENDY project. . ." reads better as "involved in the TRENDY project. . ." 6) Line 80: a comma is needed before "respectively", ". . . 56 and 44% respectively" 7) Line 101: ". . .in 2015-16 years" reads better as ". . .in years 2015-16" 8) Line 104: ". . .El Niños in 1997-98 years and 2015-16 years. . ." reads better as ". . .El Niños in years 1997-98 and 2015-16. . ." 9) Lines 119-120: Since more than one international project is listed, ". . .participated in the international carbon modelling project. . ." should read ". . .participated in international modelling projects. . ." 10) Line 123: "The detailed descriptions on its model structure. . ." reads better as "A detailed description of its model structure. . ." 11) Line 129: no space is needed before the comma after the reference in ". . .Anglia Climatic Research Unit et al., 2014) , NOAA's. . ." 12) Lines 149-150: Capitalize the expansion of the MACC acronym (e.g., ". . .Atmospheric Composition & Climate. . ." 13) Line 168: Unit (K) is needed for temperature anomaly of 2.0 14) Line 168: "El Niño event tends to. . ." reads better as "An El Niño event tends to. . ." 15) Line 170: "growth rate" should be plural, "growth rates" 16) Line 173: Remove extraneous period after Mount. 17) Line 173: ". . .during 1982-83 El Niño event" reads better as ". . .during the 1982-83 El Niño event" 18) Line 315: "...tropics, opposing to composite and. . ." reads better as ". . .tropics, as opposed to the composite and. . ." 19) Line 325: ". . .anomalously higher, opposing to the cooler during. . ." reads better as ". . .anomalously higher, as opposed to the cooling during. . ." 20) Line 331: ". . .more attentions have been paid on SIF.." reads better as ". . .more attention has been paid to SIF" 21) Line 338: ". . .increased over America, Southern South America. . .". The location needs to be better described. Perhaps change, "America" to "North America". 22) Line 339: ". . .but decreases" should be changed to past tense like the rest of the sentence, ". . .but decreased" 23) Lines 340-341: ". . .anomalies were well corresponding to simulated. . ." reads better as ". . .anomalies corresponded well to simulated. . ." 24) Line 344: "add a comma after "disturbances for FTA," 25) Line 346: "Globally" should be lowercase

26) Line 390: "...El Niño episode, opposing to GPP..." reads better as "...El Niño episode, as opposed to GPP..." 27) Line 393: The word "the" is not needed in the phrase "air temperature over the North America" 28) Lines 395-396: "...higher, opposing the cooler in..." reads better as "...higher, as opposed to the cooling in..." 29) Line 400: "the" is needed in the phrase "...frequently happening in the tropics" 30) Line 456: A period is needed after the reference for consistency 31) Line 539: Randerson et al. reference does not follow alphabetical order. It should be moved before Schwalm in line 531. 32) Line 583: "a It represents..." the word "It" is not needed 33) Line 593: MLO should be defined in the caption like the other acronyms are 34) Line 607: "And the arrows" reads better as "The arrows" 35) Line 609: "And the purple" reads better as "The purple" 36) Line 609: "denotes result" reads better as "denotes the result" 37) Line 613: the lat/lon coordinates for extratropical NH and tropics should be defined in the caption so the reader doesn't have to skim through the text when looking at the figure. 38) Line 622: the lat/lon coordinates for extratropical NH and tropics should be defined in the caption so the reader doesn't have to skim through the text when looking at the figure. 39) Line 635: Figure 6 colorbar values are too small to read. Perhaps, include only 1 larger bar for each variable on the figure, rather than 3 small colorbars.

---

## Short Comment (SC1) · 31 Jul 2017

General Comments: The paper dedicated to the different terrestrial carbon cycle responses to the two strongest El Niño events. It is clear that two El Niño events accompany discriminable anomalies in terrestrial carbon flux and it may due to the different spatial pattern in soil wetness and air temperature anomalies.

Wang et al. is well written, the analysis and methodology well described, and the results will no doubt be of interest to the readers of Earth System Dynamics after some additional work.

[Figure]

Specific Comments: 1. Statistical significance In Fig. 3 to 6, the composite results are shown by averaging anomalies for eight El Niño events, except for 1982-83 and 1991-92. However, authors did not show significance levels, so it is hard to say common features in El Niño events. In addition, two extreme cases have the larger anomalies than composite results, but it is needed that how significant between extreme cases and composite results as normal cases. By using bootstrap estimation, it can be possible to address P-value and significant levels.

Then, it would be more clear that how anomalies in soil wetness and air temperature act regional terrestrial carbon flux, especially for two extreme El Niño events.

2. Seasonal evolution Recently, Kim et al. (2016) argued that carbon flux in South Asia lead to the delayed peak in the ENSO-related carbon cycle. Authors already analysed regionally, but more detail analysis, as like Kim et al. (2016), is needed in order to understand different features in the delayed peak for two extreme El Niño events.

Technical Corrections: 1. Line 24 and 373: El Nino -> El Niño

Reference Kim, J.-S., Kug, J.-S., Yoon, J.-H. & Jeong, S.-J. Increased atmospheric $CO_2$ growth rate during El Niño driven by reduced terrestrial productivity in the CMIP5 ESMs. J. Clim. 29, 8783-8805 (2016).

---

## Referee Comment (RC2) · Anonymous Referee #2 · 9 Aug 2017

The response of terrestrial carbon cycle to ENSO has been a hot topic for terrestrial carbon cycle community for a long time. Most of the earlier studies focus on the general responses built on an ensemble of ENSO events. However, it is clear that each ENSO is different, and therefore, their resulting response from the terrestrial ecosystems is expected to differ. Yet, such event-based case study is lacking in literature is due to the lack of appropriate data constraints. Thus I believe that Wang et al. paper has the potential to complement current literature.

But my major concern regarding this paper is the data constrains they applied. The authors need to confirm their readers that atmospheric CO2 growth rate can provide

constraint on a single event, and on small regional scales. The authors have shown that VEGAS is highly correlated with atmospheric CO2 growth rate, however, this does not ensure that VEGAS can capture net CO2 flux anomalies from a single event. For example, a recent study on ERL by Fang et al. found that mechanistic models can capture ENSO response fairly well when all years are considered, however, they all have some issues when considering only El Nino or La Nina years. It is ok to use VEGAS to explore the driving mechanisms; however, some caveats are needed.

I agree with the other reviewer that statistical significance tests for anomalies, composites etc are needed, which may help strengthen the paper (i.e., Figure 2,3,4 etc).

I also agree with the other reviewer that it would be good to check whether seasonal evolution of climatic drivers, GPP and Respiration matter.

My other comment is about the fire emissions. The authors mentioned that FTA anomaly is 1.95 Pg C per yr during 1997-1998, while is 0.8 Pg C per yr during 2015-2016 (that is, 1.1 Pg C per yr difference between two events). In their paper, they showed that the difference of fire emission of CO2 from GFED is 0.82 Pg C per yr between these two events, so fire emissions only can explain 70% of the difference between two ENSO events, is this correct? Is it fair to conclude that fire emission dominates the difference and thus explore why fire emission differs in the paper?

Detailed comments: 1. abstract: seems to be too long, and has two paragraphs. Better to shorten it. 2. I wonder if "two strongest El Nino events" used in the title and throughout the paper is appropriate. First, two strongest events are defined only since 1980, right? So it is not in history. Second, how to define how strong an El Nino is depends on which aspects you talked about. I would probably just use two strong El Nino events or two extreme El Nino events instead to make the statement more accurate. 3. Explain somewhere early in the paper that positive sign of the cartbon fluxes discussed here means to the atmosphere. 4. Introduction: There are actually more observation-based studies that argue temperature is more important driver. While many of the paper cited

here in Line 78 are mostly model-based results, and models have be shown to over-estimate the role of precipitation (see, Piao et al., 2013 and Fang et al. 2017) . 5. Introduction: line 86, here "sensitivity analysis" is not the right word and is misleading for this paper (wang et al., 2013), I think this number is the slope based on regression analysis. 6. Results: Line 184-185: it is true that models can capture the general response to ENSO with a moderate correlation coefficient. However, a recent ERL study shows they have problem in capturing response to El Nino vs Response to La Nina. 7. Results: line 196-197, why use the mean of CAMs and MACC? 8. Figure 2c and 3d, why there appears to be two strong peaks for the inversion?

References: 1. Piao et al 2013 http://onlinelibrary.wiley.com/doi/10.1111/gcb.12187/ 2. Fang et al. 2017 http://iopscience.iop.org/article/10.1088/1748-9326/aa6e8e/

―――――――――――――――――

---

## Author Comment (AC1) · 3 Sep 2017

**Responses to Referee #1 comment on "Contrasting terrestrial carbon cycle responses to the two strongest El Niño events: 1997–98 and 2015–16 El Niños"**

Dear Referee and Editor,

Thank you very much for your efforts to deal with our manuscript and provide constructive comments. We have tried our best to re-summarize the results, and modify this manuscript accordingly. The following is our point-by-point reply to the comments.

1) Introduction: While the literature review is comprehensive and the introduction clearly describes the problem and the state of the science, the novelty of this research needs to be more clearly stated in the introduction. I suggest including a sentence explicitly stating how this research is novel compared to previous studies up front so the reader can better understand how this research is set apart from other studies.

Reply: Thanks very much for your suggestions. We have added a sentence "Therefore, it is of great importance for us to have the clear insight into the impacts of ENSO events on the terrestrial carbon cycle through the typical case study." in the introduction to illustrate the importance of the comparison in the impacts between 1997-98 and 2015-16 El Ninos.

2) Conclusions and Discussion: The conclusions are clearly outlined and are consistent with the interpretation of the results. However, this section seems to be more conclusion, and is lacking in discussion. This left me interested with many questions that should be added after the conclusions, such as the caveats of this study (model, datasets, etc.), implications of the research (i.e., how does this research advance our science), and what, if any, future research may be done to build on the conclusions established (i.e., additional model/data analysis, additional El Niño years analyzed, etc.). More discussion would tie the manuscript and the state of the science in better, and will give a better big picture view.

Reply: Thanks very much for your suggestions. We have added some discussions after conclusions according to your suggestions. Part of them is as below: "*In addition, we*

*should be cautious that the responses of terrestrial carbon cycle to El Niño events in this study are simulated by an individual DGVM, namely VEGAS. Uncertainties remain among the different state-of-the-art DGVMs owing to their different model structures, biological processes considered, parameterizations, and so on (Piao et al., 2013; Sitch et al., 2015; Wang et al., 2016). If possible, we can quantify the inter-model uncertainties in regional responses of terrestrial carbon cycle to El Niño events when the new round TRENDY simulations (TRENDY-v6, 1901–2016) are available. Also, though we take three inversion datasets as references for the VEGAS simulation in the context, they cover different periods. Importantly, there are also large uncertainties among different atmospheric $CO_2$ inversions because of their different prescribed priors, a priori uncertainties, inverse methods, and observational datasets (Peylin et al., 2013). Future atmospheric $CO_2$ inversions may give us more accurate results based on more observational datasets including the surface and satellite-based observations. ...".* Details can be seen in the context.

References:

(1) Peylin, P., Law, R. M., Gurney, K. R., Chevallier, F., Jacobson, A. R., Maki, T., Niwa, Y., Patra, P. K., Peters, W., Rayner, P. J., Rödenbeck, C., van der Laan-Luijkx, I. T., and Zhang, X.: Global atmospheric carbon budget: results from an ensemble of atmospheric $CO_2$ inversions, Biogeosciences, 10, 6699-6720, 2013.

(2) Piao, S., Sitch, S., Ciais, P., Friedlingstein, P., Peylin, P., Wang, X., Ahlström, A., Anav, A., Canadell, J. G., Cong, N., Huntingford, C., Jung, M., Levis, S., Levy, P. E., Li, J., Lin, X., Lomas, M. R., Lu, M., Luo, Y., Ma, Y., Myneni, R. B., Poulter, B., Sun, Z., Wang, T., Viovy, N., Zaehle, S., and Zeng, N.: Evaluation of terrestrial carbon cycle models for their response to climate variability and to $CO_2$ trends, Global Change Biology, doi: 10.1111/gcb.12187, 2013. 2117–2132, 2013.

(3) Sitch, S., Friedlingstein, P., Gruber, N., Jones, S. D., Murray-Tortarolo, G., Ahlström, A., Doney, S. C., Graven, H., Heinze, C., Huntingford, C., Levis, S., Levy, P. E.,

Lomas, M., Poulter, B., Viovy, N., Zaehle, S., Zeng, N., Arneth, A., Bonan, G., Bopp, L., Canadell, J. G., Chevallier, F., Ciais, P., Ellis, R., Gloor, M., Peylin, P., Piao, S. L., Le Quéré, C., Smith, B., Zhu, Z., and Myneni, R.: Recent trends and drivers of regional sources and sinks of carbon dioxide, Biogeosciences, 12, 653-679, 2015.

(4) Wang, J., Zeng, N., and Wang, M.: Interannual variability of the atmospheric $CO_2$ growth rate: roles of precipitation and temperature, Biogeosciences, 13, 2339-2352, 2016.

**Technical Corrections:**

1) Line 16: It is not clear what CO2 variability is being addressed. Perhaps, specify "The large interannual atmospheric CO2 variability. . ."

Reply: Thanks very much. We have modified it accordingly.

2) Line 21: Same comment as above, "Mauna Loa atmospheric CO2 concentration. . ."

Reply: Thanks very much. We have modified it.

3) Line 42: ". . .opposing to the cooler in. . ." would read better as "opposing the cooling in. . ."

Reply: Thanks very much. We have modified.

4) Line 68: for consis- tency and clarity, the variable "Cfire" should have a written definition included like the other variables, such as "carbon flux from fire".

Reply: Thanks. We have added the definition of "Cfire" according to your suggestion in the context.

5) Line 73: ". . .involved in TRENDY project. . ." reads better as "involved in the TRENDY project. . ."

Reply: Thanks for your suggestion. We have modified.

6) Line 80: a comma is needed before "respectively", ". . . 56 and 44% respectively"

Reply: Thanks very much. We have modified.

7) Line 101: ". . .in 2015-16 years" reads better as ". . .in years 2015-16"

Reply: Thanks very much. We have modified.

8) Line 104: ". . .El Niños in 1997-98 years and 2015-16 years. . ." reads better as ". . .El Niños in years 1997-98 and 2015-16. . ."

Reply: Thanks very much. We have modified.

9) Lines 119-120: Since more than one international project is listed, ". . .participated in the international carbon modelling project..." should read "...participated in international modelling projects. . ."

Reply: Thanks very much. We have modified.

10) Line 123: "The detailed descriptions on its model structure. . ." reads better as "A detailed description of its model structure. . ."

Reply: Thanks very much. We have modified accordingly.

11) Line 129: no space is needed before the comma after the reference in ". . .Anglia Climatic Research Unit et al., 2014) , NOAA's. . ."

Reply: Thanks very much. We have modified accordingly.

12) Lines 149-150: Capitalize the expansion of the MACC acronym (e.g., ". . .Atmospheric Composition & Climate. . ."

Reply: Thanks very much. We have modified accordingly.

13) Line 168: Unit (K) is needed for temperature anomaly of 2.0

Reply: Thanks very much. We have modified accordingly.

14) Line 168: "El Niño event tends to. . ." reads better as "An El Niño event tends to. . ."

Reply: Thanks very much. We have modified accordingly.

15) Line 170: "growth rate" should be plural, "growth rates"

Reply: Thanks very much. We have modified accordingly.

16) Line 173: Remove extraneous period after Mount.

Reply: Thanks very much. We have modified accordingly.

17) Line 173: "...during 1982-83 El Niño event" reads better as "...during the 1982-83 El Niño event"

Reply: Thanks very much. We have modified accordingly.

18) Line 315: "...tropics, opposing to composite and. . ." reads better as "...tropics, as opposed to the composite and..."

Reply: Thanks very much. We have modified accordingly.

19) Line 325: "...anomalously higher, opposing to the cooler during..." reads better as "...anomalously higher, as opposed to the cooling during..."

Reply: Thanks very much. We have modified accordingly.

20) Line 331: "...more attentions have been paid on SIF.." reads better as "...more attention has been paid to SIF"

Reply: Thanks very much. We have modified accordingly.

21) Line 338: "...increased over America, Southern South America...". The location needs to be better described. Perhaps change, "America" to "North America".

Reply: Thanks very much. We have modified accordingly.

22) Line 339: ". . .but decreases" should be changed to past tense like the rest of the

sentence, ". . .but de- creased"

Reply: Thanks very much. We have modified accordingly.

23) Lines 340-341: ". . .anomalies were well corresponding to simulated. . ." reads better as ". . .anomalies corresponded well to simulated. . ."

Reply: Thanks very much. We have modified accordingly.

24) Line 344: "add a comma after "disturbances for FTA,"

Reply: Thanks very much. We have modified accordingly.

25) Line 346: "Globally" should be lowercase

Reply: Thanks very much. We have modified accordingly.

26) Line 390: "...El Niño episode, opposing to GPP..." reads better as "...El Niño episode, as opposed to GPP. . ."

Reply: Thanks very much. We have modified accordingly.

27) Line 393: The word "the" is not needed in the phrase "air temperature over the North America"

Reply: Thanks very much. We have modified accordingly.

28) Lines 395-396: ". . .higher, oppos- ing the cooler in. . ." reads better as ". . .higher, as opposed to the cooling in. . ."

Reply: Thanks very much. We have modified accordingly.

29) Line 400: "the" is needed in the phrase ". . .frequently happening in the tropics"

Reply: Thanks very much. We have modified accordingly.

30) Line 456: A period is needed after the reference for consistency

Reply: Thanks very much. We have modified accordingly.

31) Line 539: Randerson et al. reference does not follow alphabetical order. It should be moved before Schwalm in line 531.

Reply: Thanks very much. We have modified accordingly.

32) Line 583: "a It represents. . ." the word "It" is not needed

Reply: Thanks very much. We have modified accordingly.

33) Line 593: MLO should be defined in the caption like the other acronyms are

Reply: Thanks very much. We have modified accordingly.

34) Line 607: "And the arrows" reads better as "The arrows"

Reply: Thanks very much. We have modified accordingly.

35) Line 609: "And the purple" reads better as "The purple"

Reply: Thanks very much. We have modified accordingly.

36) Line 609: "denotes result" reads better as "denotes the result"

Reply: Thanks very much. We have modified accordingly.

37) Line 613: the lat/lon coordinates for extratropical NH and tropics should be defined in the caption so the reader doesn't have to skim through the text when looking at the figure.

Reply: Thanks very much. We have modified accordingly.

38) Line 622: the lat/lon coordinates for extratropical NH and tropics should be defined in the caption so the reader doesn't have to skim through the text when looking at the figure.

Reply: Thanks very much. We have modified accordingly.

39) Line 635: Figure 6 colorbar values are too small to read. Perhaps, include only 1 larger bar for each variable on the figure, rather than 3 small colorbars.

Reply: Thanks for your suggestions. We have tried our best to zoom in the colorbars. It looks better now.

---

## Author Comment (AC2) · 4 Sep 2017

**Responses to Referee #2 comment on "Contrasting terrestrial carbon cycle responses to the two strongest El Niño events: 1997–98 and 2015–16 El Niños" Dear Referee and Editor,**

Thank you very much for your efforts to deal with our manuscript and provide constructive comments. We have tried our best to re-summarize the results, and modify this manuscript accordingly. The following is our point-by-point reply to the comments.

(1) But my major concern regarding this paper is the data constrains they applied. The authors need to confirm their readers that atmospheric CO2 growth rate can provide constraint on a single event, and on small regional scales. The authors have shown that VEGAS is highly correlated with atmospheric CO2 growth rate, however, this does not ensure that VEGAS can capture net CO2 flux anomalies from a single event. For example, a recent study on ERL by Fang et al. found that mechanistic models can capture ENSO response fairly well when all years are considered, however, they all have some issues when considering only El Nino or La Nina years. It is ok to use VEGAS to explore the driving mechanisms; however, some caveats are needed.

Reply: Thanks very much for your suggestions. I totally agree with you that there are biases in all of the state-of-the-art model simulations (Piao et al., 2013; Sitch et al., 2015; Wang et al., 2016). Also, the atmospheric CO2 growth rate indeed cannot provide any constraint on regional scales. So we take some recent datasets including three inversions (MACC, CAMS, and CarbonTracker) and satellite-based observations (EVI and SIF) as references for spatial simulations by VEGAS. Of course, uncertainties exist among inversion datasets because of their different prescribed priors, a priori uncertainties, inverse methods, and observational datasets selected (Peylin et al, 2013). Maybe future inversions can give us more accurate results with the increased surface and satellite-based CO2 observations. Accordingly, we have added some discussions after the concluding remarks to inform readers that model and datasets used all have biases (or uncertainties). There is still a long road to improve DGVMs in modelling community.

- (2) I agree with the other reviewer that statistical significance tests for anomalies, composites etc are needed, which may help strengthen the paper (i.e., Figure 2,3,4 etc). Reply: Thanks very much for your suggestions. We have made the statistical significance tests for composite anomalies based on the bootstrap estimation. You can see them in the modified paper.
- (3) I also agree with the other reviewer that it would be good to check whether seasonal evolution of climatic drivers, GPP and Respiration matter. Reply: Thanks very much. In this paper, we mainly focus on the contrasting responses of terrestrial carbon cycle to the two extreme El Ninos (1997/98 and 2015/16) during the whole El Nino period. Also, we covered some information of seasonal evolutions in total C flux anomaly section (seen in Figure 2-4). The spatial seasonal evolutions during the El Nino events are also a good topic. Actually, we also want to present the seasonal evolutions during the 2015/16 El Nino with temperature and precipitation regional contributions by model sensitivity experiments in another paper.
- (4) My other comment is about the fire emissions. The authors mentioned that FTA anomaly is 1.95 Pg C per yr during 1997-1998, while is 0.8 Pg C per yr during 2015-2016 (that is, 1.1 Pg C per yr difference between two events). In their paper, they showed that the difference of fire emission of CO2 from GFED is 0.82 Pg C per yr between these two events, so fire emissions only can explain 70% of the difference between two ENSO events, is this correct? Is it fair to conclude that fire emission dominates the difference and thus explore why fire emission differs in the paper? Reply: Thanks very much. But I disagree with you.

First, according to  $\delta F_{TA} \cong \delta TER - \delta GPP + \delta C_{fire}$ , we can get  $\delta F_{TA}$ =1.14 Pg C yr-1,  $\delta TER$ =-1.14 Pg C yr-1,  $\delta GPP$ =-1.9 Pg C yr-1, and  $\delta C_{fire}$ =0.38 Pg C yr-1 between 1997/98 and 2015/16 El Ninos simulated by VEGAS, respectively. So FTA difference between two events is largely determined by differences in TER and GPP. Of course, fire emissions simulated by VEGAS was underestimated in 1997/98 (Table 2).

Second, GFED fire emission datasets used here only covers the period from 1997 through 2014 (Randerson et al., 2015). So we only have the Cfire anomaly with the value of 0.82 Pg C yr-1 in 1997/98 without the values in 2015/16. We cannot say "the difference of fire emission of CO2 from GFED is 0.82 Pg C per yr between these two events". So It is wrong that fire emissions can explain 70% of the difference between two ENSO events. We need more up-to-date observations to quantify the difference in fire emissions between two extreme El Ninos.

**Detailed comments:**

- abstract: seems to be too long, and has two paragraphs. Better to shorten it.
   Reply: Thanks for your suggestions. We have tried our best to make the abstract clear and concise.
- (2) I wonder if "two strongest El Nino events" used in the title and through- out the paper is appropriate. First, two strongest events are defined only since 1980, right? So it is not in history. Second, how to define how strong an El Nino is depends on which aspects you talked about. I would probably just use two strong El Nino events or two extreme El Nino events instead to make the statement more accurate. Reply: Thanks for your constructive suggestions. We have modified "two strongest El Nino events" into "the extreme El Nino events" throughout the paper.
- (3) Explain somewhere early in the paper that positive sign of the cartbon fluxes discussed here means to the atmosphere.
  Reply: Thanks for your suggestions. We have added this information in the second paragraph in Introduction as follows "Directly, land-atmosphere C flux (FTA, positive sign is into the atmosphere) is mainly attributable to the imbalance between the gross primary productivity (GPP) and terrestrial ecosystem respiration (TER)..."
- (4) Introduction: There are actually more observation-based studies that argue temperature is more important driver. While many of the paper cited here in Line 78

are mostly model-based results, and models have be shown to over- estimate the role of precipitation (see, Piao et al., 2013 and Fang et al. 2017) Reply: Thanks very much for your suggestions. We have added some paper such as Clark et al., 2003, Doughty et al., 2008 in Introduction to illustrate the observationbased evidence for temperature dominance.

(5) Introduction: line 86, here "sensitivity analysis" is not the right word and is misleading for this paper (wang et al., 2013), I think this number is the slope based on regression analysis.
Reply: Thanks very much. We have modified "sensitivity analysis" into "regression

analysis" according to your suggestions.

- (6) Results: Line 184-185: it is true that models can capture the general re- sponse to ENSO with a moderate correlation coefficient. However, a recent ERL study shows they have problem in capturing response to El Nino vs Response to La Nina. Reply: DGVM models can well capture the response to ENSO with significant correlation coefficients (In this paper and Figure 5 in Wang et al., 2016) in long time series on interannual time scales. We also agree that there are biases in certain El Nino or La Nina event, about which we have added some discussions. We also added Fang et al. (2017) study result in the discussion to inform that state-of-the-art DGVMs may still have some problem in capturing response to El Nino vs Response to La Nina. In this paper, we also used three inversion results as references for VEGAS simulations. The spatial anomaly of FTA in VEGAS in 2015/16 is consistent with that in CarbonTracker. This consistency gives us some confidence in model simulation results.
- (7) Results: line 196-197, why use the mean of CAMs and MACC?

Reply: These two inversion datasets (CAMS and MACC, Chevallier, 2013) have similar results on the interannual time scales (Figure 1). So we take the mean of them as one reference dataset in the study.

- (8) Figure 2c and 3d, why there appears to be two strong peaks for the inversion?
  - Reply: It's a good question. Comparing Figure 2c and 3d, we can know the two peaks mainly come from the tropical anomalies. We here present evolution of the spatial anomalies in CAMS and MACC during 1997/98 (Figure R.1). We can clearly see that strong positive anomalies occurred over the Indonesia, South Asia, Africa, part of Amazon, and Southern South America in tropics during the two peak periods (Aug-Oct 1997 and Mar-May 1998). In contrast, strong negative anomalies occurred over Southern Africa and Southern South America during the low period (Nov 1997 to Feb 1998).

Figure R.1. FTA evolutions in CAMS and MACC during 1997/98 El Nino.

**Reference:**

- (1) Chevallier, F.: On the parallelization of atmospheric inversions of CO2 surface fluxes within a variational framework, Geosci Model Dev, 6, 783-790, 2013.
- (2) Clark, D. A., Piper, S. C., Keeling, C. D., and Clark, D. B.: Tropical rain forest tree growth and atmospheric carbon dynamics linked to internnual tempreature variation

during 1984-2000, P. Natl. Acad. Sci. USA, 100, 5852-5857, 2003.

- (3) Doughty, C. E., and Goulden, M. L.: Are tropical forests near a high temperature threshold?, J. Geophys. Res., 113, G00B07, 2008.
- (4) Fang, Y., Michalak, A. M., Schwalm, C. R., Huntzinger, D. N., Berry, J. A., Ciais, P., Piao, S. L., Poulter, B., Fisher, J. B., Cook, R. B., Hayes, D., Huang, M. Y., Ito, A., Jain, A., Lei, H. M., Lu, C. Q., Mao, J. F., Parazoo, N. C., Peng, S. S., Ricciuto, D. M., Shi, X. Y., Tao, B., Tian, H. Q., Wang, W. L., Wei, Y. X., and Yang, J.: Global land carbon sink response to temperature and precipitation varies with ENSO phase, Environ. Res. Lett., 12, 064007, 2017.
- (5) Peylin, P., Law, R. M., Gurney, K. R., Chevallier, F., Jacobson, A. R., Maki, T., Niwa, Y., Patra, P. K., Peters, W., Rayner, P. J., Rödenbeck, C., van der Laan-Luijkx, I. T., and Zhang, X.: Global atmospheric carbon budget: results from an ensemble of atmospheric CO2 inversions, Biogeosciences, 10, 6699-6720, 2013.
- (6) Piao, S., Sitch, S., Ciais, P., Friedlingstein, P., Peylin, P., Wang, X., Ahlström, A., Anav, A., Canadell, J. G., Cong, N., Huntingford, C., Jung, M., Levis, S., Levy, P. E., Li, J., Lin, X., Lomas, M. R., Lu, M., Luo, Y., Ma, Y., Myneni, R. B., Poulter, B., Sun, Z., Wang, T., Viovy, N., Zaehle, S., and Zeng, N.: Evaluation of terrestrial carbon cycle models for their response to climate variability and to CO2 trends, Global Change Biology, doi: 10.1111/gcb.12187, 2013. 2117–2132, 2013.
- (7) Randerson, J. T., van der Werf, G. R., Giglio, L., Collatz, G. J. and Kasibhatla, P. S.:Global Fire Emissions Database, Version 4, (GFEDv4). ORNL DAAC, Oak Ridge, Tennessee, USA. http://dx.doi.org/10.3334/ORNLDAAC/1293, 2015.
- (8) Sitch, S., Friedlingstein, P., Gruber, N., Jones, S. D., Murray-Tortarolo, G., Ahlström, A., Doney, S. C., Graven, H., Heinze, C., Huntingford, C., Levis, S., Levy, P. E., Lomas, M., Poulter, B., Viovy, N., Zaehle, S., Zeng, N., Arneth, A., Bonan, G., Bopp, L., Canadell, J. G., Chevallier, F., Ciais, P., Ellis, R., Gloor, M., Peylin,

P., Piao, S. L., Le Quéré, C., Smith, B., Zhu, Z., and Myneni, R.: Recent trends and drivers of regional sources and sinks of carbon dioxide, Biogeosciences, 12, 653-679, 2015.

(9) Wang, J., Zeng, N., and Wang, M.: Interannual variability of the atmospheric CO2 growth rate: roles of precipitation and temperature, Biogeosciences, 13, 2339-2352, 2016.

---

## Author Comment (AC3) · 4 Sep 2017

The comment was uploaded in the form of a supplement:
https://www.earth-syst-dynam-discuss.net/esd-2017-46/esd-2017-46-AC3-supplement.pdf

---

## Author Comment (AC4) · 5 Sep 2017

**Response to J.-S. Kim comment on "Contrasting terrestrial carbon cycle responses to the two strongest El Niño events: 1997–98 and 2015–16 El Niños"**

Dear Kim,

Thank you very much for your constructive comments. We have tried our best to re-summarize the results, and modify this manuscript accordingly. The following is our point-by-point reply to the comments.

(1) Statistical significance In Fig. 3 to 6, the composite results are shown by averaging anomalies for eight El Niño events, except for 1982-83 and 1991- 92. However, authors did not show significance levels, so it is hard to say common features in El Niño events. In addition, two extreme cases have the larger anomalies than composite results, but it is needed that how significant between extreme cases and composite results as normal cases. By using bootstrap estimation, it can be possible to address P-value and significant levels. Then, it would be more clearer that how anomalies in soil wetness and air temperature act regional terrestrial carbon flux, especially for two extreme El Niño events.

Reply: Thanks very much for your suggestions. Statistical significance is indeed needed for the composite analysis. So we have added the statistical significant test in Figs. 3 to 6 in the modified paper through the popular bootstrap estimation.

(2) Seasonal evolution Recently, Kim et al. (2016) argued that carbon flux in South Asia lead to the delayed peak in the ENSO-related carbon cycle. Authors already analysed regionally, but more detail analysis, as like Kim et al. (2016), is needed in order to understand different features in the delayed peak for two extreme El Niño events.

Reply: Thanks very much for your suggestions. Actually, we also suggested that the ENSO-related carbon cycle had the delayed relationship (Qian et al., 2008; Wang et al., 2016). Seasonal evolution during the El Nino events is also a good topic. In this paper, we covered some information of seasonal evolutions in total C flux anomaly section (seen in Figure 2-4). Actually, we also want to present the spatial

seasonal evolutions during the 2015/16 El Nino with temperature and precipitation regional contributions by model sensitivity experiments in another paper.

Technical Corrections:

1. Line 24 and 373: El Nino -> El Niño

   Reply: Thanks very much for your corrections. We have modified them accordingly.

---

## Author Response (AR1)

**Responses to the comments on "Contrasting terrestrial carbon cycle responses to the two strongest El Niño events: 1997–98 and 2015–16 El Niños"**

Dear Referees and Editor,

Thank you very much for your efforts to deal with our manuscript and provide constructive comments. We have tried our best to re-summarize the results, and modify this manuscript accordingly. We also have our manuscript polished by the native English-speaking expert. The following is our point-by-point reply to the comments.

**Reply to Referee #1**

1) Introduction: While the literature review is comprehensive and the introduction clearly describes the problem and the state of the science, the novelty of this research needs to be more clearly stated in the introduction. I suggest including a sentence explicitly stating how this research is novel compared to previous studies up front so the reader can better understand how this research is set apart from other studies.

Reply: Thanks very much for your suggestions. We have added a sentence "Therefore, it is important to have clear insight into the impacts of ENSO events on the terrestrial carbon cycle, and this is best achieved through representative case studies." in the introduction to illustrate the importance of the comparison in the impacts between 1997/98 and 2015/16 El Nino events.

2) Conclusions and Discussion: The conclusions are clearly outlined and are consistent with the interpretation of the results. However, this section seems to be more conclusion, and is lacking in discussion. This left me interested with many questions that should be added after the conclusions, such as the caveats of this study (model, datasets, etc.), implications of the research (i.e., how does this research advance our science), and what, if any, future research may be done to build on the conclusions established (i.e., additional model/data analysis, additional El Niño years analyzed, etc.). More discussion would tie the manuscript and the state of the science in better, and will give a better big picture view.

Reply: Thanks very much for your suggestions. We have added some discussions after conclusions according to your suggestions. Part of them is as below: "*It is important to keep in mind that the responses of the terrestrial carbon cycle to the El Niño events in this study were simulated using an individual DGVM (VEGAS), which, whilst highly consistent with the variations in the CGR and inversion results, carries uncertainties in terms of the regional responses because of, for example, its model structure, biological processes considered, and parameterizations. Of course, uncertainties exist in all of the state-of-the-art DGVMs. Fang et al. (2017) recently suggested that none of the 10 contemporary terrestrial biosphere models captures the ENSO-phase-dependent responses. If possible, we will quantify the inter-model uncertainties in regional responses of the terrestrial carbon cycle to El Niño events when the new round of TRENDY simulations (1901–2016) becomes available. Although we used three inversion datasets as reference for the VEGAS simulation in this study, they cover different periods. Importantly, there are also large uncertainties between the different atmospheric $CO_2$ inversions because of their different prescribed priors, a priori uncertainties, inverse methods, and observational datasets (Peylin et al., 2013). Future atmospheric $CO_2$ inversions may produce more accurate results based on more observational datasets, including surface and satellite-based observations. ...".* Details can be seen in the context.

References:

(1) Peylin, P., Law, R. M., Gurney, K. R., Chevallier, F., Jacobson, A. R., Maki, T., Niwa, Y., Patra, P. K., Peters, W., Rayner, P. J., Rödenbeck, C., van der Laan-Luijkx, I. T., and Zhang, X.: Global atmospheric carbon budget: results from an ensemble of atmospheric $CO_2$ inversions, Biogeosciences, 10, 6699-6720, 2013.

(2) Fang, Y., Michalak, A. M., Schwalm, C. R., Huntzinger, D. N., Berry, J. A., Ciais,

P., Piao, S. L., Poulter, B., Fisher, J. B., Cook, R. B., Hayes, D., Huang, M. Y., Ito,

A., Jain, A., Lei, H. M., Lu, C. Q., Mao, J. F., Parazoo, N. C., Peng, S. S., Ricciuto,

D. M., Shi, X. Y., Tao, B., Tian, H. Q., Wang, W. L., Wei, Y. X., and Yang, J.:

Global land carbon sink response to temperature and precipitation varies with

ENSO phase, Environ. Res. Lett., 12, 064007, 2017.

**Technical Corrections:**

1) Line 16: It is not clear what CO2 variability is being addressed. Perhaps, specify

"The large interannual atmospheric CO2 variability. . ."

Reply: Thanks very much. We have modified it accordingly.

2) Line 21: Same comment as above, "Mauna Loa atmospheric CO2 concentration. . ."

Reply: Thanks very much. We have modified it.

3) Line 42: ". . .opposing to the cooler in. . ." would read better as "opposing the cooling in. . ."

Reply: Thanks very much. We have modified.

4) Line 68: for consis- tency and clarity, the variable "Cfire" should have a written definition included like the other variables, such as "carbon flux from fire".

Reply: Thanks. We have added the definition of "Cfire" according to your suggestion in the context.

5) Line 73: ". . .involved in TRENDY project. . ." reads better as "involved in the

TRENDY project. . ."

Reply: Thanks for your suggestion. We have modified.

6) Line 80: a comma is needed before "respectively", ". . . 56 and 44% respectively"

Reply: Thanks very much. We have modified.

7) Line 101: ". . .in 2015-16 years" reads better as ". . .in years 2015-16"

Reply: Thanks very much. We have modified.

8) Line 104: ". . .El Niños in 1997-98 years and 2015-16 years. . ." reads better as ". . .El

Niños in years 1997-98 and 2015-16. . ."

Reply: Thanks very much. We have modified.

9) Lines 119-120: Since more than one international project is listed, ". . .participated in the international carbon modelling project..." should read "...participated in inter- national modelling projects. . ."

Reply: Thanks very much. We have modified.

10) Line 123: "The detailed descriptions on its model structure. . ." reads better as "A

detailed description of its model structure. . ."

Reply: Thanks very much. We have modified accordingly.

11) Line 129: no space is needed before the comma after the reference in ". . .Anglia

Climatic Research Unit et al., 2014) , NOAA's. . ."

Reply: Thanks very much. We have modified accordingly.

12) Lines 149-150: Capitalize the expansion of the MACC acronym (e.g.,

". . .Atmospheric Composition & Climate. . ."

Reply: Thanks very much. We have modified accordingly.

13) Line 168: Unit (K) is needed for temperature anomaly of 2.0

Reply: Thanks very much. We have modified accordingly.

14) Line 168: "El Niño event tends to. . ." reads better as "An El Niño event tends to. . ."

Reply: Thanks very much. We have modified accordingly.

15) Line 170: "growth rate" should be plural, "growth rates"

Reply: Thanks very much. We have modified accordingly.

16) Line 173: Remove extraneous period after Mount.

Reply: Thanks very much. We have modified accordingly.

17) Line 173: "...during 1982-83 El Niño event" reads better as "...during the 1982-83

El Niño event"

Reply: Thanks very much. We have modified accordingly.

18) Line 315: "...tropics, opposing to composite and. . ." reads better as "...tropics, as opposed to the composite and..."

Reply: Thanks very much. We have modified accordingly.

19) Line 325: "...anomalously higher, opposing to the cooler during..." reads better as

"...anomalously higher, as opposed to the cooling during..."

Reply: Thanks very much. We have modified accordingly.

20) Line 331: "...more attentions have been paid on SIF.." reads better as "...more attention has been paid to SIF"

Reply: Thanks very much. We have modified accordingly.

21) Line 338: "...increased over America, Southern South America...". The location needs to be better described. Perhaps change, "America" to "North America".

Reply: Thanks very much. We have modified accordingly.

22) Line 339: ". . .but decreases" should be changed to past tense like the rest of the sentence, ". . .but de- creased"

Reply: Thanks very much. We have modified accordingly.

23) Lines 340-341: ". . .anomalies were well corresponding to simulated. . ." reads better as ". . .anomalies corresponded well to simulated. . ."

Reply: Thanks very much. We have modified accordingly.

24) Line 344: "add a comma after "disturbances for FTA,"

Reply: Thanks very much. We have modified accordingly.

25) Line 346: "Globally" should be lowercase

Reply: Thanks very much. We have modified accordingly.

26) Line 390: "...El Niño episode, opposing to GPP..." reads better as "...El Niño episode, as opposed to GPP. . ."

Reply: Thanks very much. We have modified accordingly.

27) Line 393: The word "the" is not needed in the phrase "air temperature over the

North America"

Reply: Thanks very much. We have modified accordingly.

28) Lines 395-396: ". . .higher, oppos- ing the cooler in. . ." reads better as ". . .higher, as opposed to the cooling in. . ."

Reply: Thanks very much. We have modified accordingly.

29) Line 400: "the" is needed in the phrase ". . .frequently happening in the tropics"

Reply: Thanks very much. We have modified accordingly.

30) Line 456: A period is needed after the reference for consistency

Reply: Thanks very much. We have modified accordingly.

31) Line 539: Randerson et al. reference does not follow alphabetical order. It should be moved before Schwalm in line 531.

Reply: Thanks very much. We have modified accordingly.

32) Line 583: "a It represents. . ." the word "It" is not needed

Reply: Thanks very much. We have modified accordingly.

33) Line 593: MLO should be defined in the caption like the other acronyms are

Reply: Thanks very much. We have modified accordingly.

34) Line 607: "And the arrows" reads better as "The arrows"

Reply: Thanks very much. We have modified accordingly.

35) Line 609: "And the purple" reads better as "The purple"

Reply: Thanks very much. We have modified accordingly.

36) Line 609: "denotes result" reads better as "denotes the result"

Reply: Thanks very much. We have modified accordingly.

37) Line 613: the lat/lon coordinates for extratropical NH and tropics should be defined in the caption so the reader doesn't have to skim through the text when looking at the figure.

Reply: Thanks very much. We have modified accordingly.

38) Line 622: the lat/lon coordinates for extratropical NH and tropics should be defined in the caption so the reader doesn't have to skim through the text when looking at the figure.

Reply: Thanks very much. We have modified accordingly.

39) Line 635: Figure 6 colorbar values are too small to read. Perhaps, include only 1

larger bar for each variable on the figure, rather than 3 small colorbars.

Reply: Thanks for your suggestions. We have tried our best to zoom in the colorbars. It looks better now.

**Reply to Referee #2**

(1) But my major concern regarding this paper is the data constrains they applied. The authors need to confirm their readers that atmospheric $CO_2$ growth rate can provide constraint on a single event, and on small regional scales. The authors have shown that VEGAS is highly correlated with atmospheric $CO_2$ growth rate, however, this does not ensure that VEGAS can capture net $CO_2$ flux anomalies from a single event. For example, a recent study on ERL by Fang et al. found that mechanistic models can capture ENSO response fairly well when all years are considered, however, they all have some issues when considering only El Nino or La Nina years.

It is ok to use VEGAS to explore the driving mechanisms; however, some caveats are needed.

Reply: Thanks very much for your suggestions. I totally agree with you that there are biases in all of the state-of-the-art model simulations (Piao et al., 2013; Sitch et al.,

2015; Wang et al., 2016). Also, the atmospheric $CO_2$ growth rate indeed cannot provide any constraint on regional scales. So we take some recent datasets including three inversions (MACC, CAMS, and CarbonTracker) and satellite-based observations (EVI

and SIF) as reference for spatial simulations by VEGAS. Of course, uncertainties exist among inversion datasets because of their different prescribed priors, a priori uncertainties, inverse methods, and observational datasets selected (Peylin et al, 2013). Maybe future inversions can give us more accurate results with the increased surface and satellite-based CO2 observations. Accordingly, we have added some discussions after the concluding remarks to inform readers that model and datasets used all have biases (or uncertainties). There is still a long road to improve DGVMs in modelling community.

(2) I agree with the other reviewer that statistical significance tests for anomalies, composites etc are needed, which may help strengthen the paper (i.e., Figure 2,3,4 etc).

Reply: Thanks very much for your suggestions. We have made the statistical significance tests for composite anomalies based on the bootstrap estimation and Student's $t$-test. You can see them in the modified paper.

(3) I also agree with the other reviewer that it would be good to check whether seasonal evolution of climatic drivers, GPP and Respiration matter.

Reply: Thanks very much. In this paper, we mainly focus on the contrasting responses of terrestrial carbon cycle to the two extreme El Ninos (1997/98 and 2015/16) during the whole El Nino period. Also, we covered some information of seasonal evolutions in total C flux anomaly section (seen in Figure 2-4). The spatial seasonal evolutions during the El Nino events are also a good topic. Actually, we also want to present the seasonal evolutions during the 2015/16 El Nino with temperature and precipitation regional contributions by model sensitivity experiments in another paper.

(4) My other comment is about the fire emissions. The authors mentioned that FTA

anomaly is 1.95 Pg C per yr during 1997-1998, while is 0.8 Pg C per yr during

2015- 2016 (that is, 1.1 Pg C per yr difference between two events). In their paper, they showed that the difference of fire emission of CO2 from GFED is 0.82 Pg C

per yr between these two events, so fire emissions only can explain 70% of the difference between two ENSO events, is this correct? Is it fair to conclude that fire emission dominates the difference and thus explore why fire emission differs in the paper?

Reply: Thanks very much. But I disagree with you.

First, according to $\delta F_{TA} \cong \delta TER - \delta GPP + \delta C_{fire}$, we can get $\delta F_{TA}$=1.14 Pg C yr$^{-1}$,

$\delta TER$=$-1.14$ Pg C yr$^{-1}$, $\delta GPP$=$-1.9$ Pg C yr$^{-1}$, and $\delta C_{fire}$=0.38 Pg C yr$^{-1}$ between

1997/98 and 2015/16 El Ninos simulated by VEGAS, respectively. So $F_{TA}$ difference between two events is largely determined by differences in TER and GPP. Of course, fire emissions simulated by VEGAS was underestimated in 1997/98 (Table 2).

Second, GFED fire emission datasets used here only covers the period from 1997

through 2014 (Randerson et al., 2015). So we only have the Cfire anomaly with the value of 0.82 Pg C yr$^{-1}$ in 1997/98 without the values in 2015/16. We cannot say "the difference of fire emission of CO2 from GFED is 0.82 Pg C per yr between these two events". So It is wrong that fire emissions can explain 70% of the difference between two ENSO events. We need more up-to-date observations to quantify the difference in fire emissions between two extreme El Ninos.

**Detailed comments:**

(1) abstract: seems to be too long, and has two paragraphs. Better to shorten it.

Reply: Thanks for your suggestions. We have tried our best to make the abstract clear and concise.

(2) I wonder if "two strongest El Nino events" used in the title and through- out the paper is appropriate. First, two strongest events are defined only since 1980, right?

So it is not in history. Second, how to define how strong an El Nino is depends on which aspects you talked about. I would probably just use two strong El Nino events or two extreme El Nino events instead to make the statement more accurate.

Reply: Thanks for your constructive suggestions. We have modified "two strongest El

Nino events" into "the extreme El Nino events" throughout the paper.

(3) Explain somewhere early in the paper that positive sign of the cartbon fluxes discussed here means to the atmosphere.

Reply: Thanks for your suggestions. We have added this information in the second paragraph in Introduction as follows "Directly, land-atmosphere C flux ($F_{TA}$, positive sign meaning a flux into the atmosphere) is mainly attributable to the imbalance between the gross primary productivity (GPP) and terrestrial ecosystem respiration (TER)…"

(4) Introduction: There are actually more observation-based studies that argue temperature is more important driver. While many of the paper cited here in Line

78 are mostly model-based results, and models have be shown to over- estimate the role of precipitation (see, Piao et al., 2013 and Fang et al. 2017)

Reply: Thanks very much for your suggestions. We have added some paper such as

Clark et al., 2003, Doughty et al., 2008 in Introduction to illustrate the observation- based evidence for temperature dominance.

(5) Introduction: line 86, here "sensitivity analysis" is not the right word and is misleading for this paper (wang et al., 2013), I think this number is the slope based on regression analysis.

Reply: Thanks very much. We have modified "sensitivity analysis" into "regression analysis" according to your suggestions.

(6) Results: Line 184-185: it is true that models can capture the general re- sponse to

ENSO with a moderate correlation coefficient. However, a recent ERL study shows they have problem in capturing response to El Nino vs Response to La

Nina.

Reply: Thanks very much. DGVM models can well capture the response to ENSO with significant correlation coefficients (In this paper and Figure 5 in Wang et al., 2016) in long time series on interannual time scales. We also agree that there are biases in certain

El Nino or La Nina event, about which we have added some discussions. We also added

Fang et al. (2017) study result in the discussion to inform that state-of-the-art DGVMs may still have some problem in capturing response to El Nino vs Response to La Nina.

In this paper, we also used three inversion results as references for VEGAS simulations.

The spatial anomaly of $F_{TA}$ in VEGAS in 2015/16 is consistent with that in

CarbonTracker. This consistency gives us some confidence in model simulation results.

(7) Results: line 196-197, why use the mean of CAMs and MACC?

Reply: Thanks very much. These two inversion datasets (CAMS and MACC,

Chevallier, 2013) have similar results on the interannual time scales (Figure 1). So we take the mean of them as one reference dataset in the study.

(8) Figure 2c and 3d, why there appears to be two strong peaks for the inversion?

Reply: It's a good question. Comparing Figure 2c and 3d, we can know the two peaks mainly come from the tropical anomalies. We here present evolution of the spatial anomalies in CAMS and MACC during 1997/98 (Figure R.1). We can clearly see that strong positive anomalies occurred over the Indonesia, South Asia, Africa, part of

Amazon, and Southern South America in tropics during the two peak periods (Aug-Oct and Mar-May 1998). In contrast, strong negative anomalies occurred over southern Africa and southern South America during the low period (Nov 1997 to Feb 1998).

[Figure]

Figure R.1. $F_{TA}$ evolutions in CAMS and MACC during 1997/98 El Nino.

[revised manuscript text omitted]


[revised manuscript text omitted]

---

## Author Response (AR2)

**Responses to the comments on "Contrasting terrestrial carbon cycle responses to the 1997/98 and 2015/16 extreme El Niño events"**

Dear Referees and Editor,

Thank you very much for your efforts to deal with our manuscript and provide constructive comments. We have tried our best to re-summarize the results, and modify this manuscript accordingly. The following is our point-by-point reply to the comments.

**Reply to Referee #2**

The authors did a good job in revising their manuscript following the comments from the reviewers. The only suggestion I would have is corresponding to their revision - We have added a sentence. "Therefore, it is important to have clear insight into the impacts of ENSO events on the terrestrial carbon cycle, and this is best achieved through representative case studies." I feel that the first half of the sentence does not justify what this study is specially designed for, and is too general. I would suggest add "individual' between the "the impacts of" and "ENSO events".

In climate science, scientists have published review paper (A. Capotondi, A. T. Wittenberg, M. Newman, E. Di Lorenzo, J.-Y. Yu, P. Braconnot, J. Cole, B. Dewitte, B. Giese, E. Guilyardi, F.-F. Jin, K. Karnauskas, B. Kirtman, T. Lee, N. Schneider, Y. Xue, and S.-W. Yeh. Bulletin of the American Meteorological Society, 2015. DOI: 10.1175/BAMS-D-13-00117.1) on ENSO diversity and pointed out that each ENSO event is different from the other. I think what this review suggests clearly indicates the value and importance of studying how Carbon cycle responds to individual ENSO

events.

I would suggest accept with small amendments.

**Reply:** Thanks very much for your constructive suggestions. We have added the "individual" between the "the impacts of" and "ENSO events" in the context. Also, we have added this reference (Capotondi et al., 2015) into the text for illustrating the differences in  $F_{TA}/CO_2$  CGR responses to different ENSO events, seen as "
[revised manuscript text omitted]